# Loss Function Learning for Domain Generalization by Implicit Gradient

## Abstract

Generalising robustly to distribution shift is a major challenge that is pervasive across most real-world applications of machine learning. A recent study highlighted that many advanced algorithms proposed to tackle such domain generalisation (DG) fail to outperform a properly tuned empirical risk minimisation (ERM) baseline. We take a different approach, and explore the impact of the ERM loss function on out-of-domain generalisation. In particular, we introduce a novel meta-learning approach to loss function search based on implicit gradient. This enables us to discover a general purpose parametric loss function that provides a drop-in replacement for cross-entropy. Our loss can be used in standard training pipelines to efficiently train robust models using any neural architecture on new datasets. The results show that it clearly surpasses cross-entropy, enables simple ERM to outperform some more complicated prior DG methods, and provides state-of-the-art performance across a variety of DG benchmarks. Furthermore, unlike most existing DG approaches, our setup applies to the most practical setting of single-source domain generalisation, on which we show significant improvement.

## 1 Introduction

Deep learning is highly successful when the training and testing samples meet the i.i.d. assumption. However, this assumption is violated in many practical applications of machine learning from medical imaging to earth observation imaging (Koh et al., 2021). This has led a large number of studies to investigate approaches to training models with increased robustness to distribution shift at testing-time, a problem setting known as Domain Generalisation (DG). Despite the volume of research in this area (Zhou et al., 2021a), a recent careful benchmarking exercise, DomainBed (Gulrajani & Lopez-Paz, 2021) showed that simple empirical risk minimisation (ERM) on a combination of training domains is a very strong baseline when properly tuned. State-of-the-art alternatives based on sophisticated architectures, regularisers, and data augmentation schemes failed to reliably beat ERM (Gulrajani & Lopez-Paz, 2021).

Rather than propose an alternative to ERM for DG, we investigate a previously unstudied hyper-parameter of ERM, namely the choice of loss function—which has been ubiquitously taken to be standard cross-entropy (CE) in prior DG work. Loss function choice has been shown to impact calibration (Mukhoti et al., 2020), overfitting (Gonzalez & Miikkulainen, 2019), and label-noise robustness (Wang et al., 2019) in standard supervised learning, so it is intuitive that it would impact robustness to domain-shift. However, it has not yet been studied in this context. Our preliminary experiments showed that equipping ERM with some recent robust loss functions in place of CE does lead to improvements in DG performance where sophisticated alternatives have failed (Gulrajani & Lopez-Paz, 2021). This raises the question: can one design a loss function specialised for DG?

To answer this question, we define a meta-learning algorithm to learn a parametric (white-box) loss function suitable for DG. Our desiderata are: (1) Performing ERM with this loss on a source domain should lead to good performance when tested on out-of-domain target data; and (2) It should provide a 'plug-and-play' drop-in replacement for cross-entropy that, once learned, can be used without further modification or computational expense with any new dataset or model architecture. While there has been growing interest in meta-learning for loss function design (Li et al., 2019a), they mostly fail to meet these criteria. They learn problem-specific—rather than re-usable—losses. If applied to DG, this would imply replacing simple ERM learning with sophisticated meta-learning

pipelines to train a loss on a per-problem basis. In contrast, our discovered loss provides a drop in replacement for CE that leads standard training pipelines to produce more robust models.

To train a general purpose robust loss function we need a search space that is flexible enough to include interesting new losses, but simple enough to generalise across tasks without overfitting to the problem used for loss learning. We choose a 12-dimensional space of fourth order Taylor polynomials. Furthermore, we need a loss that is suitable for all stages of training. This precludes the majority of approaches based on online meta-learning which update the loss and base model iteratively (Li et al., 2019a;c), and also suffer from short-horizon bias (Wu et al., 2018). Evolutionary methods (Gonzalez & Miikkulainen, 2019) and reinforcement-learning (Li et al., 2019a) could support loss learning in principle, but are too slow to be feasible. Therefore we develop the first implicit-gradient based approach to loss learning. This allows us to tractably compute meta-gradients of the target recognition performance with respect to the loss used for training in the source domain.

We use a simple DG task (RotatedMNIST) to train our robust loss, termed Implicit Taylor Loss (ITL), to replace CE in ERM. Subsequent experiments show that ERM with ITL surpasses CE across a range of DG benchmarks, and leads to state of the art performance, despite being much simpler and faster than competitor DG methods. While the majority of existing DG methods require multiple source domains to conduct data augmentation or feature alignment strategies, ITL improves *single-source* domain generalisation, a crucial problem setting which has been minimally studied thus far.

To summarise our contributions: (i) We provide the first study on the significance of supervised loss function choice in DG (ii) We demonstrate the first efficient solution to loss-learning based on meta-gradients computed by the Implicit Function Theorem. (iii) Empirically, we show that our learned ITL loss enhances simple ERM and achieves state of the art DG performance across a range of benchmarks, including the challenging single-source DG scenario.

## 2 RELATED WORK

**Domain Generalisation:** Domain Generalisation aims to learn a model using data from one or more source domains, but with the further requirement that it is robust to testing on novel target domain data—without accessing target data during training. DG is now a well studied (Zhou et al., 2021a) area with diverse approaches including data augmentation (Shankar et al., 2018; Zhou et al., 2021b), robust training algorithms such as domain alignment objectives (Li et al., 2018b), and other regularisers (Li et al., 2019c; Balaji et al., 2018). Most DG studies have assumed the *multi-source* setting, which enables new data-augmentation strategies (Zhou et al., 2021b), and allows generalisation-promoting design features to be tuned by domain-wise cross-validation. In particular, a few studies (Li et al., 2019c; Balaji et al., 2018) have considered meta-learning based DG, where a regulariser applied in a training domain is tuned by meta-gradients from the resulting validation-domain performance. The resulting model is then deployed to the true target domain within the same family. These methods require regulariser meta-learning for each given multi-source DG problem family. In contrast, we propose to learn a simple loss function once, which then provides a drop-in replacement for CE in any single-, or multi-source DG problem. A recent criticism of the DG literature showed that no method consistently outperformed a well tuned ERM baseline on the carefully designed DomainBed benchmark (Gulrajani & Lopez-Paz, 2021). Rather than competing with ERM, we simply enhance the ERM loss function and this leads to a clear improvement on DomainBed.

**Loss Function learning:** Loss Function learning aims to discover new losses that improve model optimisation from various perspectives including conventional generalisation performance, (Gonzalez & Miikkulainen, 2019; Liu et al., 2021), optimisation efficiency (Li et al., 2019a; Gonzalez & Miikkulainen, 2019; Wang et al., 2020; Bechtle et al., 2020), and noise robustness (Li et al., 2019a). Key dichotomies are in the search space of black box (neural) (Bechtle et al., 2020; Li et al., 2019c) vs white-box (human-readable) (Li et al., 2019a; Gonzalez & Miikkulainen, 2019; Wang et al., 2020) losses; whether learned losses are problem specific (Li et al., 2019a; Wang et al., 2020) or reusable (Gonzalez & Miikkulainen, 2019); the meta-optimisation algorithm used—evolution (Liu et al., 2021; Gonzalez & Miikkulainen, 2019), RL (Li et al., 2019a; Wang et al., 2020; Bechtle et al., 2020), or gradient (Li et al., 2019c); and whether the loss is updated offline (Liu et al., 2021; Gonzalez & Miikkulainen, 2019) (long inner loop, typically intractable), or online (Li et al., 2019a; Wang et al., 2020) (short inner loop, efficient but suffers from short-horizon bias (Wu et al., 2018)). No studies have thus far investigated loss learning for domain-shift robustness. In order to learn a reusable

robust loss we use a white-box loss search space of Taylor polynomials proposed in Gonzalez & Miikkulainen (2020), and offline/long inner loop meta-learning. To make this meta-optimisation tractable, we exploit the Implicit Function Theorem, to efficiently generate accurate hypergradients of the validation domain performance with respect to the training domain loss function parameters. Besides being the first demonstration of loss learning for DG, to our knowledge it is also the first demonstration of any implicit gradient-based loss learning.

## 3 METHOD

The need for Domain Generalisation arises when one is using machine learning to build a model where the available training data is not representative of the data that will be observed by the model once it has been deployed. In particular, it is assumed that there is an underlying distribution over domains, $P$, from which we can sample several *source* domain distributions, $\{p_1^{(s)}, ..., p_n^{(s)} \sim P\}$, to make use of during training. We can construct a training set for each of these source domain distributions by sampling $K$ data points, $D_i^{(s)} = \{(\mathbf{x}_i^{(s,j)}, y_i^{(s,j)}) \sim p_i^{(s)}\}_{j=1}^K$, and use the union of all these sets as the full training set, $D^{(s)} = \bigcup_{i=1}^n D_i^{(s)}$. Empirical Risk Minimisation (ERM) then simply finds the model parameters, $\theta$, that minimise the loss measured on this training set,

$$\min_\theta \frac{1}{n} \sum_{i=1}^n \frac{1}{K} \sum_{j=1}^K \mathcal{L}(f_\theta(\mathbf{x}_i^{(j)}), y_i^{(j)}),\tag{1}$$

where $\mathcal{L}(\cdot, \cdot)$ is a loss function (typically cross entropy) measuring how well the predicted labels match the ground truth labels. One can empirically check the resulting model's robustness to domain shift by sampling one or more *target* domain distributions, $\{p_1^{(t)}, ..., p_m^{(t)} \sim P\}$, from the same distribution over domains that was used to generate the training data. Data can then be sampled for each of these target domains, yielding a test dataset $D^{(t)} = \bigcup_{i=1}^m D_i^{(t)}$. Standard evaluation metrics such as accuracy can then be computed using this data.

### 3.1 META-LEARNING LOSSES FOR DG

Our goal is to replace the standard CE loss typically used in ERM with a learned loss function. We are motivated by recent work showing that learned losses can enable models to perform better for a variety of other problem settings, such as training with label noise (Wang et al., 2019) and improving calibration (Mukhoti et al., 2020). We formulate the task of learning the parameters, $\omega$, of a loss function, $\mathcal{L}_\omega$, as a bilevel optimisation problem. The outer objective is to find the $\omega$ that maximises the performance of a model evaluated on the target domain data, and the inner problem is to train a model to minimise the value of $\mathcal{L}_\omega$ measured on the source domain data. The loss parameters are optimised using gradient-based methods that take advantage of the implicit function theorem to efficiently compute gradients for the outer optimisation problem. Crucially, once the optimal loss function $\omega^*$ has been found, new DG problems can be solved via ERM on the $\mathcal{L}_{\omega*}$ loss.

The bilevel optimisation we use to formalise the meta-learning process is given by

$$\omega^* = \arg\min_\omega \frac{1}{m} \sum_{i=1}^m \frac{1}{K} \sum_{j=1}^K \mathcal{M}(f_{\theta^*(\omega)}(\mathbf{x}_i^{(t,j)}), y_i^{(t,j)})\tag{2}$$

$$s.t. \quad \theta^*(\omega) = \arg\min_\theta \frac{1}{n} \sum_{i=1}^n \frac{1}{K} \sum_{j=1}^K \mathcal{L}_\omega(f_\theta(\mathbf{x}_i^{(s,j)}), y_i^{(s,j)})\tag{3}$$

where $\mathcal{M}$ is a loss function used to measure the performance of the model on the target domains, typically chosen to be cross entropy.

Optimising $\omega$ is challenging due to the need to backpropagate through the long inner loop optimisation of $\theta$. Existing approaches for learning loss functions typically resort to slow evolutionary or reinforcement learning updates (Li et al., 2019a; Gonzalez & Miikkulainen, 2019; Wang et al., 2020) in the outer loop, or to an online approximation based on alternating steps on $\omega$ and $\theta$ (Li et al., 2019a; Wang et al., 2020). The latter approach leads to a loss function $\omega^*$ that cannot be transferred

to new tasks, as it suffers from a short-horizon bias (Wu et al., 2018). To solve this problem, we use the Implicit Function Theorem (IFT) to compute $\frac{\partial \mathcal{M}}{\partial \omega}$ without truncating the inner optimisation problem to approximate $\theta^*(\omega)$.

## 3.2 IMPLICIT GRADIENT

The conceptually simplest way to optimise $\omega$ is to store all the intermediate iterates generated by the optimiser when training the network in the inner loop, and to then backpropagate through all of these weight updates (Maclaurin et al., 2015). This becomes prohibitively expensive in both memory and computation. Instead, after finding $\theta^*(\omega)$ we compute the gradient using the Implicit Function Theorem (IFT). The implicit gradient computation takes advantage of the fact that $\frac{\partial \mathcal{L}_\omega}{\partial \theta} = 0$, because we have found locally optimal model parameters for the inner problem. The gradient we want to compute is given by

$$\frac{\partial \mathcal{M}}{\partial \omega} = \frac{\partial \mathcal{M}}{\partial \theta} \frac{\partial \theta}{\partial \omega} \bigg|_{\omega, \theta^*(\omega)}, \tag{4}$$

and the IFT can be used to obtain

$$\frac{\partial \theta}{\partial \omega} = -\Big[ \underbrace{\frac{\partial^2 \mathcal{L}_\omega}{\partial \theta\, \partial \theta^T}}_{|\theta| \times |\theta|} \Big]^{-1} \times \underbrace{\frac{\partial^2 \mathcal{L}_\omega}{\partial \theta\, \partial \omega^T}}_{|\theta| \times |\omega|}. \tag{5}$$

The inverse of the Hessian can be rephrased in terms of a Neumann series,

$$\Big[ \frac{\partial^2 \mathcal{L}_\omega}{\partial \theta\, \partial \theta^T} \Big]^{-1} = \lim_{i \to \infty} \sum_{j=0}^{i} [I - \frac{\partial^2 \mathcal{L}_\omega}{\partial \theta\, \partial \theta^T}]^j, \tag{6}$$

and approximated by truncating the summation to a finite number of terms. In practice, one can make use of vector-Jacobian products to avoid explicitly constructing the Hessian in the summation. Further details can be found in Lorraine et al. (2020), but we provide pseudo-code for computing the implicit gradient in Algorithm 2.

---

**Algorithm 1** IFT-based loss learning for DG.

1: **Input:** P, $\omega$
2: **Output:** $\omega^*$
3: Init $\omega$
4: **while** not converged or reached max steps **do**
5:      sample $p_1, ..., p_n$ from $P$
6:      sample $D_1, ..., D_n$ from $p_1, ..., p_n$
7:      Init $H = 0 \in \mathbb{R}^{n \times |\omega|}$
8:      **for all** $D_i$ **do**
9:          Init $\theta_i$ {Get random network weights}
10:          $D^s = \{D_1, ..., D_n\}/D_i$,   $D^t = D_i$ {Construct source/target splits}
11:          $\theta_i^* = \arg\min_\theta \mathcal{L}_\omega(\theta_i, D^s)$ {Train the network}
12:          $h_i = \text{Hypergradient}(\mathcal{L}_\omega, \mathcal{M}, (\omega, \theta_i^*), \alpha)$
13:          $H[i,:] = h_i$
14:      **end for**
15:      $h = \text{grad-surgery}(H)$
16:      $\omega = \omega - \eta h$ {Update the loss function}
17: **end while**

**Algorithm 2** Computing the hypergradient of the meta-objective $\mathcal{M}$, with respect to the loss $\omega$. The $\text{grad}(\cdot, \cdot, \cdot)$ function from PyTorch computes a Jacobian-vector product when called with a non-scalar first argument. Inspired by Lorraine et al. (2020), we use this to efficiently compute the Hessian required for approximating the Neumann series.

**Input:** $\mathcal{L}_\omega, \mathcal{M}, (\omega, \theta^*), \alpha$
**Output:** $-p\frac{\partial^2 \mathcal{L}_\omega}{\partial \theta \partial \omega}$
$v = p = \frac{\partial \mathcal{M}}{\partial \theta}\big|_{(\omega, \theta^*)}$
**for all** $j = 1, ..., J$ **do**
     $v -= \alpha \cdot \text{grad}(\frac{\partial \mathcal{L}_\omega}{\partial \theta}, \theta, v)$
     $p += v$
**end for**

---

## 3.3 ROBUST GRADIENT ESTIMATION

Algorithm 1 summarizes the gradient estimation procedure. To obtain high quality gradient estimates in each outer loop iteration, we employ a leave-one-domain-out strategy. The DG task, $P$, used for meta-training the loss parameters has $m$ domains associated with it. In each iteration of the outer loop, we train $m$ networks with the prospective loss (i.e., we instantiate $m$ different copies of the

Figure 1: Algorithm schematic. Loss $\mathcal{L}_\omega$ is trained to optimize held-out domain performance on R-MNIST and then deployed on novel datasets.

inner loop), where each network has a different target domain and the remainder of the domains are used to train the network. We can then compute a gradient for each of the $m$ networks and aggregate them together in order to perform an update to the loss parameters. Rather than using the mean gradient, we found aggregation using gradient surgery (Yu et al., 2020), which reduces the gradient noise caused by different source/target domain splits in the inner loop, to work better in practice.

### 3.4 Taylor Polynomial Representation

The choice of loss function parameterisation is a crucial factor in our framework. One must balance the ability to represent a sufficiently broad range of loss functions, with the susceptibility to overfitting the data used to learn the loss. The search space we consider is based on the truncated Taylor polynomials used by the evolutionary optimisation loss learning approach of (Gonzalez & Miikkulainen, 2020). This family of loss functions treats the point around which the Taylor polynomial is centred, and also the value of the derivatives at this point, as learnable parameters. In this sense, it is a variational learning method—though it should be stressed it is not a variational *Bayesian* method. The family of $\beta$ order multivariate Taylor polynomials has the form

$$\ell(\mathbf{z}) = \sum_{n=0}^{\beta} \frac{1}{n!} \nabla^n \ell(\mathbf{c})^T (\mathbf{z} - \mathbf{c})^n, \tag{7}$$

where $\mathbf{c}$ is a fixed point around which function is being expanded. Because $\mathbf{c}$ is fixed, the values of the derivative at this point are also fixed. As such, we can replace $c$ and $\nabla^n \ell(\mathbf{c})$ with meta-learnable parameters. This allows us to parameterize the learned loss function in terms of the gradients it should have at a meta-learnable point. We define our learnable loss as

$$\mathcal{L}_\omega(\hat{\mathbf{y}}, \mathbf{y}) = \frac{1}{C} \sum_{i=1}^{C} \ell_\omega(\hat{\mathbf{y}}_i, \mathbf{y}_i), \qquad \ell_\omega(\hat{\mathbf{y}}_i, \mathbf{y}_i) = \sum_{n=0}^{\beta} \frac{1}{n!} \nabla^n \ell([\omega_0, \omega_1])^T ([\hat{\mathbf{y}}_i, \mathbf{y}_i] - [\omega_0, \omega_1])^n, \tag{8}$$

where each $\nabla^n \ell([\omega_0, \omega_1])$ can actually be replaced by introducing more meta-parameters to $\omega$. Please see Appendix A.4 for an expanded definition of this loss function.

### 3.5 Algorithm Summary

**Meta-train:** Given a set of training domains, the loss function search space in Section 3.4, and efficient update strategy in Section 3.2 and Algorithm 1, we are able to train a robust loss function $\mathcal{L}_\omega$. We conduct such loss function learning only once using a small dataset, and then evaluate the resulting loss on a variety of larger datasets that are unseen during meta-training. **Meta-test:** Given the learned loss function $\mathcal{L}_{\omega^*}$, we fix it and use it together with the ERM algorithm for novel DG tasks. Each target problem is trained from scratch and has not been seen during loss learning. An overview of the algorithm, and the learning curve of the meta-train phase, are given in Figure 1.

## 4 Experiments

### 4.1 Dataset and Implementation Details.

**Meta-train stage:** We aim to learn a general purpose loss function that can be used in diverse DG problems, but we first need to select a dataset for initial loss learning. We chose RotatedMNIST (Ghi-

Table 1: **Resnet18** Cross-domain recognition accuracy (%) on PACS.

| Target set | Art | Cartoon | Photo | Sketch | Avg. |
|---|---|---|---|---|---|
| Epi-FCR (Li et al., 2019b) | 82.1 | 77.0 | 93.9 | 73.0 | 81.5 |
| JiGen (Carlucci et al., 2019) | 79.4 | 75.3 | 96.0 | 71.6 | 80.5 |
| MASF (Dou et al., 2019) | 80.3 | 77.2 | 95.0 | 71.7 | 81.0 |
| CrossGrad (Shankar et al., 2018) | 79.8 | 76.8 | 96.0 | 70.2 | 80.7 |
| Entropy (Zhao et al., 2020) | 80.7 | 76.4 | 96.7 | 71.8 | 81.4 |
| L2A-OT (Zhou et al., 2020) | 83.3 | 78.2 | 96.2 | 73.6 | 82.8 |
| RSC (reported in Huang et al. (2020)) | 83.43 | 80.31 | 95.99 | 80.85 | 85.15 |
| Mixstyle (rs) (Zhou et al., 2021b) | $82.3 \pm 0.2$ | $79.0 \pm 0.3$ | $96.3 \pm 0.3$ | $73.8 \pm 0.9$ | 82.8 |
| Mixstyle (dl) (Zhou et al., 2021b) | $84.1 \pm 0.4$ | $78.8 \pm 0.4$ | $96.1 \pm 0.3$ | $75.9 \pm 0.9$ | 83.7 |
| RSC (our rerun their code) | $79.25 \pm 0.69$ | $77.63 \pm 0.50$ | $93.61 \pm 0.37$ | $78.11 \pm 1.40$ | 81.91 |
| RSC + ITL | $81.67 \pm 0.80$ | $76.56 \pm 0.50$ | $95.57 \pm 0.22$ | $77.05 \pm 0.56$ | 82.71 |
| ERM + BCE | $71.19 \pm 0.81$ | $70.82 \pm 0.29$ | $93.11 \pm 0.78$ | $57.65 \pm 0.98$ | 73.19 |
| ERM + CE | $76.9 \pm 0.6$ | $76.5 \pm 0.7$ | $93.3 \pm 0.1$ | $68.8 \pm 0.6$ | 78.9 |
| ITL-Net (ERM+ITL) | $83.9 \pm 0.4$ | $78.9 \pm 0.6$ | $94.8 \pm 0.2$ | $80.1 \pm 0.6$ | **84.4** |

fary et al., 2015), which contains six different domains that are all derived from MNIST (LeCun & Cortes, 2010) but with different rotations: 0%, 15%, 30%, 45%, 60%, and 75%. The leave-one-domain-out strategy for robust implicit gradient estimation, therefore, results in six inner loop instantiations for each outer loop iteration. For efficiency, we use 2-layer MLPs as the base model, which contains 1024-256-10 units from the input layer to the output one with ReLU as the activation function. The learning rates in the inner loop and outer loop are both 0.01, a batch size of 32 is used for the inner loop, and the Neumann series used for approximating the inverse Hessian is truncated at 15 iterations. The result is a set of 12 parameters that define the learned fourth-order polynomial loss function and an example of learned loss is shown in Appendix A.3. The meta-train compute cost is described in Appendix A.6.

**Meta-test (deployment) stage:** We now evaluate our learned ITL using ERM, but with our learned loss instead of cross entropy. Models trained with our method are denoted by ITL-Net. We evaluate the learned loss function on the four common DG benchmarks: VLCS (Fang et al., 2013), PACS (Li et al., 2018a), OfficeHome (Venkateswara et al., 2017), and Terra Incognita (Beery et al., 2018). Two sets of experiments are conducted: (i) We evaluate the conventional PACS benchmark, as it is the most widely used in the DG literature, and enables comparison against the most recent state-of-the-art competitors. (ii) We evaluate all four benchmarks using the recent DomainBed platform, which is designed to enforce fair and consistent hyperparameter tuning across different methods.

## 4.2 RESULTS

**PACS: Setup** A pre-trained ResNet18 backbone is used throughout, together with the source and target domain split described in (Li et al., 2018a). We train ResNet-18 with ITL on the training split and perform model selection using the validation set. We use same set of hyperparameters (learning rate of 0.001, weight decay of 0.00001, and batch size of 32 for each domain) for the baseline ERM with Cross-Entropy (ERM +CE) and Binary Cross-Entropy (BCE) to train our model. We compare ITL-Net with the existing state-of-the-art methods on this benchmark, including RSC (Huang et al., 2020), data augmentation-based L2A-OT (Zhou et al., 2021b), Mixstyle (Zhou et al., 2020) including random shuffle (rs) and domain label (dl), regulariser-based Entropy (Zhao et al., 2020), adversarial gradient-based CrossGrad (Shankar et al., 2018), meta learning-based MASF (Dou et al., 2019) and Epi-FCR (Li et al., 2019b) and self-supervision-based JiGen (Carlucci et al., 2019).

**PACS: Results** We first conduct experiments using the classic PACS protocol to facilitate comparison against many recent competitors that were not evaluated on DomainBed. Table 1 compares our ITL-Net performance vs state-of-the-art methods. From the results we can see that: (i) Simply swapping out the loss in ERM from CE to ITL, leads to a significant 5.5% improvement. (ii) Overall our ITL-Net leads to state-of-the-art performance on this benchmark, surpassing the most recent and sophisticated competitors such as Mixstyle.

**DomainBed: Setup** We next evaluate ITL using the DomainBed platform, which enforces careful and fair evaluation by ensuring that all competitors use the same hyper-parameter tuning strategy

Table 2: **DomainBed** Cross-domain recognition accuracy (%) with **ResNet50** on **ColoredMNIST VLCS**, **PACS**, **TerraIncognita**, **OfficeHome** and **DomainNet**. Bottom: Results of Wilcoxon signed-rank hypothesis test comparing ITL-Net against competitors.

| | Models | | | | | |
|---|---|---|---|---|---|---|
| Dataset | ERM | SagNet | CORAL | CDANN | RSC | ITL-Net |
| ColoredMNIST | 51.5 | 51.7 | 51.5 | 51.7 | 51.7 | **52.0** |
| VLCS | 77.5 | 77.8 | 78.8 | 77.5 | 77.1 | **78.9** |
| PACS | 85.5 | 86.3 | 86.2 | 82.6 | 85.2 | **86.4** |
| TerraIncoginita | 46.1 | 48.6 | 47.6 | 45.8 | 46.6 | **51.0** |
| OfficeHome | 66.5 | 68.1 | 68.7 | 65.8 | 65.5 | **69.3** |
| DomainNet | 40.9 | 40.3 | 41.5 | 38.3 | 38.9 | **41.6** |
| Avg. Rank | 4.17 | 2.67 | 3.00 | 5.17 | 5.00 | 1.00 |
| p-value ($H_0$) | Reject (0.016) | Reject (0.016) | Reject (0.016) | Reject (0.016) | Reject (0.016) | |

Table 3: Cross-domain recognition accuracy (%) on **DomainBed-PACS-Resnet50**. Comparison with alternative manually-designed robust losses.

| Loss | ERM+CE | ERM+FOCAL | ERM+SCE | ERM+GCE | EMR+LS | ERM+ITL |
|---|---|---|---|---|---|---|
| Avg Perf | $83.9 \pm 0.5$ | $84.6 \pm 0.8$ | $84.2 \pm 0.5$ | $83.0 \pm 0.2$ | $84.9 \pm 0.6$ | $\mathbf{86.4} \pm 0.5$ |

(random search, driven by source domain validation performance), and the same number of hyperparameter search iterations. We follow the standard DomainBed protocol and use a ResNet-50, with experiments conducted on VLCS, PACS, OfficeHome, and Terra Incognita.

**DomainBed: Results**   The results in Table 2 compare ITL-Net with ERM and some of the most competitive published alternatives: SagNet (Nam et al., 2019), CORAL (Sun & Saenko, 2016), CDANN (Li et al., 2018c) and RSC (Huang et al., 2020) in the original DomainBed paper (Gulrajani & Lopez-Paz, 2021). A detailed comparison is given in Appendix 6. The conclusion of the DomainBed study was that existing methods did not reliably beat ERM under this hyperparameter tuning protocol. In contrast, we can see that ITL-Net provides a clear improvement on ERM and matches or improves on the strongest existing competitor in each case, especially on TerraIncognita and OfficeHome. To formally compare ITL-Net with competitors, we perform significance testing using the Wilcoxon signed-rank test, where the p-value is set as 0.025 and the sample size is number of the Domain datasets applied. For example, when comparing the performance of ITL-Net and ERM, the null hypothesis ($H_0$) is that ITL-Net has equal performance to ERM on DomainBed and the alternative hypothesis ($H_1$) that ITL-Net is statistically significantly better than ERM on DomainBed. The results of the hypothesis tests comparing ITL to each competitor are summarized at the bottom of Table 2, and confirm that ITL-Net outperforms them all.

**Single Source DG: Setup**   Most existing DG methods rely on the availability of multiple source domains in some form: For example to synthesise new domains for data augmentation (Zhou et al., 2021b), or perform feature alignment among training domains (Sun & Saenko, 2016). A unique feature of our ITL-Net is that, since it is only a small modification to ERM, it can be used to learn on a single source domain. Although this setting is not well explored in the literature, it is obviously highly practical as multiple source domains are often not available in practice. To explore this setting, we modify the DomainBed benchmark to train on a single source at a time and average over each source→target combination, rather than training on the conjunction of all sources.

**Single Source DG: Results**   From the results in Table 4, we can see that performance drops across the board compared to multi-source training (Table 2), as expected. However, the state of the art alternatives CORAL and SagNet are no longer as competitive compared to ERM as they were in the multi-source case (Table 2)—this is expected as they are designed to exploit cues from multiple source domains. In contrast, our ITL-Net maintains a clear lead over the conventional ERM with cross entropy baseline in this setting. This is a significant achievement as existing work has not produced algorithms that improve robustness under the single-source setting.

Table 4: Cross-domain recognition accuracy (%) on DomainBed with single source domain. The heading of the table denotes the single source domain, and results average across all target domains.

| Source Dataset | VLCS | PACS | OfficeHome | TerraIncognita |
|---|---|---|---|---|
| ERM | **64.08** | 51.85 | 53.57 | 32.13 |
| CORAL | 64.07 | 51.84 | 53.51 | 32.13 |
| SagNet | 61.78 | 53.00 | 51.30 | 33.93 |
| Mixup | 59.01 | 54.92 | 52.70 | 30.80 |
| ITL-Net | 62.17 | **56.54** | **55.04** | **35.09** |

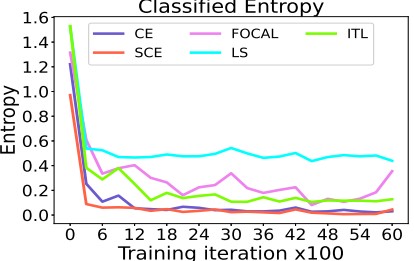 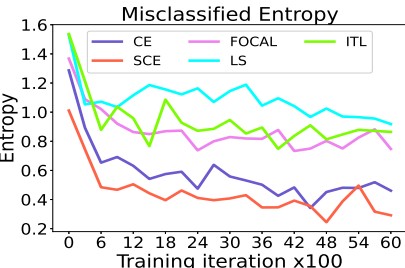

Figure 2: The evolution of posterior entropy for target domain test samples during training. Left: Correctly classified samples. Right: Misclassified samples.

### 4.3 FURTHER ANALYSIS

**Qualitative comparison and impact on learning**  To understand our learned loss, we visually compare it Figure 5 with several other loss functions: CE, SCE (Wang et al., 2019), Label Smoothing (LS) (Pereyra et al., 2017), and FOCAL (Mukhoti et al., 2020). Compared to the standard cross entropy loss, we can see that ITL has softer penalties for severe misclassification, and stronger penalties for moderate misclassification. In addition, it can be noticed that FOCAL has the most similar shape to our ITL. The properties of the FOCAL loss for probability calibration were recently studied in (Mukhoti et al., 2020). Motivated by this, we report the evolution of the target-domain entropy of correctly and incorrectly classified samples during training in Figure 2(left) and Figure 2(right) respectively. Clearly, a byproduct of ITL compared to CE and SCE is a (desirable) increase in uncertainty for misclassified instances. FOCAL has similar behaviour to ITL in general, especially the entropy of prediction distributions for misclassified instances.

**Quantitative comparison to other robust losses**  We next investigate whether the good performance of ITL in cross-domain robustness can be easily replicated by applying existing robust loss functions, or whether our meta-learning pipeline has learned something new in terms of robust model training. SCE (Wang et al., 2019) and GCE (Zhang & Sabuncu, 2018) were designed with label-noise robustness in mind, while Focal (Mukhoti et al., 2020; Lin et al., 2017) was designed for class imbalance and calibration. Label-smooth (LS) (Pereyra et al., 2017) is for improving generalisation and reducing overconfidence. From the results in Table 3, we can see that while some losses improve on CE, ITL leads to the clearest improvement. The somewhat similar behaviour of ITL and FOCAL in terms of induced entropy from Figure 2 does not carry over similar cross-domain recognition accuracy. Note that all experiments in Table 3 were run by us, while competitor performance in Table 2 is taken from (Gulrajani & Lopez-Paz, 2021).

**Loss landscape analysis and Perturbation analysis**  One of the factors that affect model generalisation is the loss landscape at convergence. (Keskar et al., 2017; Chaudhari et al., 2019) observed that flatter loss landscapes lead to good generalisation of the learned model. To this end, we compared a 1D slice through the loss landscape of ITL-Net with that of ERM on both source domain and target domains. Namely, we perturb the converged paraeters by moving it around though gradient direction which generated by the evigenvector of the Hessian matrix . From Figure 4, we can see that for each held out target domain, ITL-Nets have flatter loss landscapes compared with models trained by CE with respect to the source domains. To further analyse the quality of the minimas provided by CE

Table 5: Cross-domain recognition accuracy (%) on **OfficeHome**: Impact of meta-train seed (± standard deviation), and choice of pre-training dataset.

| Target set | Artistic | Clipart | Product | Real World | Avg. |
|---|---|---|---|---|---|
| ITL-NET (RotatedMNIST) | $64.22 \pm 0.84$ | $56.25 \pm 0.38$ | $77.52 \pm 0.32$ | $78.12 \pm 0.32$ | $69.03 \pm 0.18$ |
| ITL-NET (RotatedKMNIST) | $64.28 \pm 0.30$ | $55.84 \pm 0.29$ | $76.89 \pm 1.04$ | $77.96 \pm 0.30$ | $68.74 \pm 0.35$ |

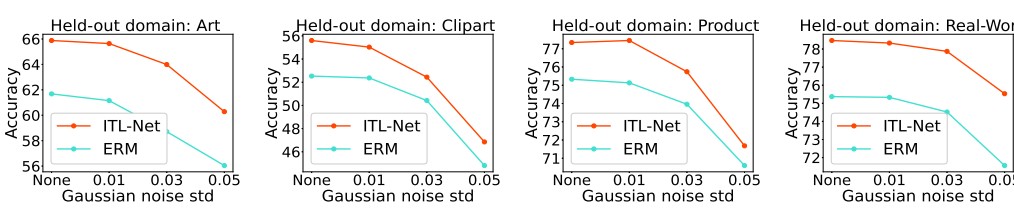

Figure 3: Perturbation analysis on OfficeHome: ITL-Net vs ERM. Multiplicative Gaussian noise with mean 1 and std: 0.01, 0.05, 0.08 is added to network weights.

and ITL, we follow the perturbation analysis routine in (Keskar et al., 2017; Zhang et al., 2018) by adding multiplicative noise to the weights of the converged models. From the results in Figure 3, it can be see that ITL-Nets outperform ERM at every noise ratio.

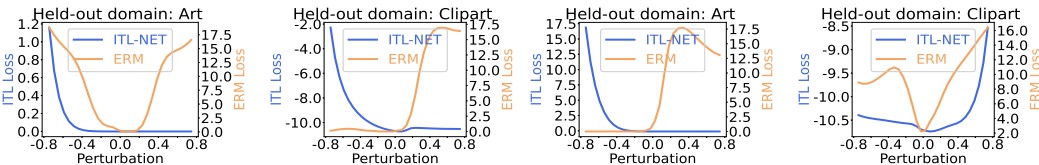

Figure 4: 1D Loss Landscape: ITL-Net vs ERM on OfficeHome. Left two: Source domain loss landscape. Right two: Target domain loss landscape.

**Repeatability analysis** Our message thus far is that a single loss produced by our pipeline can be re-used as a plug-and-play modification to improve vanilla ERM+CE on a wide variety of held-out downstream DG tasks. That said, one might reasonably wonder about the reliability of the loss function learning procedure itself. To investigate this, we repeat our entire pipeline *including* the meta-train stage five times. We then evaluate the consistency of the resulting five loss functions on the downstream ColoredMNIST task. Furthermore, to evaluate the dependence of our result on the choice of meta-training dataset, we repeat the above experiment on RotatedKMNIST (Clanuwat et al., 2018) to replace the RotatedMNIST used previously. From the results in Table 5, we can see that performance is quite consistent over trials (small standard deviation). It also differs little with choice of pre-training dataset - with both options performing well compared to competitors in Table 2.

**Limitations** ITL improves ERM+CE for DG tasks in general, but in some cases, the margin over SOTA is small, since other state-of-the-art competitors may beat ERM+CE. It would be more interesting if ITL is highly complementary to SOTA methods based on other architectural, augmentation, or domain-alignment improvements – but this remains for future work to determine.

## 5 CONCLUSION

We provided the first study of the effect of ERM loss functions on Domain Generalisation. We observe that models trained by ERM with existing robust loss functions can improve performance on Domain Generalisation compared with those trained by Cross-Entropy. To discover the best loss for DG, we perform meta-learning to find a re-usable white-box loss function. This is tractably solved using IFT to obtain gradients of the target domain performance with respect to the source domain loss parameters. This also provides the first demonstration of IFT-based loss learning in the literature. The results show that a simple modification to the standard ERM pipeline improves both multi-source and single source DG, and even surpasses the purpose-designed state of the art models.

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

# A  APPENDIX

## A.1  THE DETAILED RESULTS FOR DOMAINBED

In Table 2 of the main paper, we summarized performance across held out target domains for the standard multi-source DG problem. In Table 6 we now give the detailed results of ITL-Net for each target domain.

Table 6: **DomainBed** Cross-domain recognition accuracy (%) with **ResNet50** on **ColoredMNIST VLCS**, **PACS**, **TerraIncognita**, **OfficeHome** and **DomainNet**.

| ColoredMNIST | Target set | +90% | +80% | -90% | Avg. |
|---|---|---|---|---|---|
| | ERM | $71.7 \pm 0.1$ | $72.9 \pm 0.2$ | $10.0 \pm 0.1$ | 51.5 |
| | SagNet | $71.8 \pm 0.2$ | $73.0 \pm 0.2$ | $10.3 \pm 0.0$ | 51.7 |
| | CORAL | $71.6 \pm 0.3$ | $73.1 \pm 0.1$ | $9.9 \pm 0.1$ | 51.5 |
| | CDANN | $72.0 \pm 0.3$ | $73.0 \pm 0.2$ | $10.2 \pm 0.1$ | 51.7 |
| | RSC | $71.9 \pm 0.3$ | $72.9 \pm 0.4$ | $10.2 \pm 0.2$ | 51.8 |
| | ITL-Net | $71.3 \pm 0.6$ | $73.4 \pm 0.1$ | $11.3 \pm 0.7$ | **52.0** |

| VLCS | Target set | Caltech | Labelme | Sun | V-Pascal | Avg. |
|---|---|---|---|---|---|---|
| | ERM | $97.7 \pm 0.4$ | $64.3 \pm 0.9$ | $73.4 \pm 0.5$ | $77.3 \pm 1.3$ | 77.5 |
| | SagNet | $97.9 \pm 0.4$ | $64.5 \pm 0.5$ | $71.4 \pm 1.3$ | $77.5 \pm 0.5$ | 77.8 |
| | CORAL | $98.3 \pm 0.1$ | $66.1 \pm 1.2$ | $73.4 \pm 0.3$ | $77.5 \pm 1.2$ | 78.8 |
| | CDANN | $97.3 \pm 0.3$ | $65.1 \pm 1.2$ | $70.7 \pm 0.8$ | $77.1 \pm 1.5$ | 77.5 |
| | RSC | $97.9 \pm 0.1$ | $62.5 \pm 0.7$ | $72.3 \pm 1.2$ | $75.6 \pm 0.8$ | 77.1 |
| | ITL-Net | $98.3 \pm 0.4$ | $65.4 \pm 0.7$ | $75.1 \pm 0.6$ | $76.8 \pm 1.2$ | **78.9** |

| PACS | Target set | Art | Cartoon | Photo | Sketch | Avg. |
|---|---|---|---|---|---|---|
| | ERM | $84.7 \pm 0.4$ | $80.8 \pm 0.6$ | $97.2 \pm 0.3$ | $79.3 \pm 1.0$ | 85.5 |
| | SagNet | $87.4 \pm 1.0$ | $80.7 \pm 0.6$ | $97.1 \pm 0.1$ | $80.0 \pm 0.4$ | 86.3 |
| | CORAL | $88.3 \pm 0.2$ | $80.0 \pm 0.5$ | $97.5 \pm 0.3$ | $78.8 \pm 1.3$ | 86.2 |
| | CDANN | $84.6 \pm 1.8$ | $75.5 \pm 0.9$ | $96.8 \pm 0.3$ | $73.5 \pm 0.6$ | 82.6 |
| | RSC | $85.4 \pm 0.8$ | $79.1 \pm 0.6$ | $96.9 \pm 0.5$ | $77.7 \pm 1.7$ | 84.9 |
| | ITL-Net | $87.1 \pm 0.4$ | $83.3 \pm 0.6$ | $96.1 \pm 0.4$ | $79.3 \pm 0.6$ | **86.4** |

| TerraIncognita | Target set | L100 | L38 | L43 | L46 | Avg. |
|---|---|---|---|---|---|---|
| | ERM | $49.8 \pm 4.4$ | $42.1 \pm 1.4$ | $56.9 \pm 1.8$ | $35.7 \pm 3.9$ | 46.1 |
| | SagNet | $53.0 \pm 2.9$ | $43.0 \pm 2.5$ | $57.9 \pm 0.6$ | $40.4 \pm 1.3$ | 48.6 |
| | CORAL | $51.6 \pm 2.4$ | $42.2 \pm 1.0$ | $57.0 \pm 1.0$ | $39.8 \pm 2.9$ | 47.6 |
| | CDANN | $47.0 \pm 1.9$ | $41.3 \pm 4.8$ | $54.9 \pm 1.7$ | $39.8 \pm 0.8$ | 45.8 |
| | RSC | $50.2 \pm 2.2$ | $39.2 \pm 1.4$ | $56.3 \pm 1.4$ | $40.8 \pm 0.6$ | 46.6 |
| | ITL-Net | $58.4 \pm 3.7$ | $46.2 \pm 1.8$ | $58.5 \pm 0.9$ | $40.9 \pm 1.8$ | **51.0** |

| OfficeHome | Target set | Artistic | Clipart | Product | Real World | Avg. |
|---|---|---|---|---|---|---|
| | ERM | $61.3 \pm 0.7$ | $52.4 \pm 0.3$ | $75.8 \pm 0.1$ | $76.6 \pm 0.3$ | 66.5 |
| | SagNet | $63.4 \pm 0.2$ | $54.8 \pm 0.4$ | $75.8 \pm 0.4$ | $78.3 \pm 0.3$ | 68.1 |
| | CORAL | $65.3 \pm 0.4$ | $54.4 \pm 0.5$ | $76.5 \pm 0.1$ | $78.4 \pm 0.5$ | 68.7 |
| | CDANN | $61.0 \pm 1.4$ | $50.4 \pm 2.4$ | $74.4 \pm 0.9$ | $76.6 \pm 0.8$ | 65.8 |
| | RSC | $60.7 \pm 1.4$ | $51.4 \pm 0.3$ | $74.8 \pm 1.1$ | $75.1 \pm 1.3$ | 65.5 |
| | ITL-Net | $65.6 \pm 0.4$ | $55.6 \pm 0.4$ | $77.5 \pm 0.3$ | $78.6 \pm 0.4$ | **69.3** |

| DomainNet | Target set | Clipart | Infograph | Painting | Quickdraw | Real | Sketch | Avg. |
|---|---|---|---|---|---|---|---|---|
| | ERM | $58.1 \pm 0.3$ | $18.8 \pm 0.3$ | $46.7 \pm 0.3$ | $12.2 \pm 0.4$ | $59.6 \pm 0.1$ | $49.8 \pm 0.4$ | 40.9 |
| | SagNet | $57.7 \pm 0.3$ | $19.0 \pm 0.2$ | $45.3 \pm 0.3$ | $12.7 \pm 0.5$ | $58.1 \pm 0.5$ | $48.8 \pm 0.2$ | 40.3 |
| | CORAL | $59.2 \pm 0.1$ | $19.7 \pm 0.2$ | $46.6 \pm 0.3$ | $13.4 \pm 0.4$ | $59.8 \pm 0.2$ | $50.1 \pm 0.6$ | 41.5 |
| | CDANN | $54.6 \pm 0.4$ | $17.3 \pm 0.1$ | $43.7 \pm 0.9$ | $12.1 \pm 0.7$ | $56.2 \pm 0.4$ | $45.9 \pm 0.5$ | 38.3 |
| | RSC | $55.0 \pm 1.2$ | $18.3 \pm 0.5$ | $44.4 \pm 0.6$ | $12.2 \pm 0.2$ | $55.7 \pm 0.7$ | $47.8 \pm 0.9$ | 38.9 |
| | ITL-Net | $63.5 \pm 0.3$ | $19.4 \pm 0.1$ | $46.3 \pm 0.1$ | $13.7 \pm 0.4$ | $53.2 \pm 0.6$ | $53.5 \pm 0.3$ | **41.6** |

| Competitors | ERM | SagNet | CORAL | RSC | CDANN |
|---|---|---|---|---|---|
| $H_0$ (p-value) | Reject (0.016) | Reject (0.016) | Reject (0.016) | Reject (0.016) | Reject (0.016) |

## A.2 DETAILED RESULTS FOR SINGLE SOURCE DOMAIN EXPERIMENT

In Table 4 of the main paper, we reported single-source DG results, summarising performance across choice of source domain and over all target domains. In Table 7 we now give the detailed results of ITL-Net for each choice of source and target domain.

Table 7: **DomainBed** Single source domain recognition accuracy (%) with **ResNet50** on **VLCS**, **PACS**, **TerraIncognita** and **OfficeHome**. Each cell reports the accuracy for a set of target domains, and the source domain used for training corresponding to the column. The performance of target domains is separated by '/'. Average over target domains for a given source domain is given at the bottom of the cell.

| | Source set | Caltech | Labelme | Sun | V-Pascal | Avg. |
|---|---|---|---|---|---|---|
| VLCS | ERM | 47.81/55.58/59.72 54.37 | 70.00/57.61/65.90 64.5 | 52.93/62.27/60.30 58.5 | 96.75/63.29/76.81 78.95 | **64.08** |
| | CORAL | 47.81/55.58/59.72 54.37 | 70.00/57.61/65.90 64.5 | 52.93/62.27/60.30 58.5 | 96.75/63.29/76.81 78.95 | **64.08** |
| | SagNet | 48.76/53.14/56.93 52.94 | 35.69/53.53/63.33 50.85 | 67.28/62.05/66.70 65.34 | 96.96/62.88/75.20 78.35 | 61.87 |
| | ITL-Net | 43.19/39.85/51.01 44.68 | 89.82/55.18/60.63 68.54 | 48.90/61.78/60.01 56.9 | 97.31/60.84/77.51 78.55 | 62.17 |
| | Source set | Art | Cartoon | Photo | Sketch | Avg. |
| PACS | ERM | 65.36/96.23/45.41 69.0 | 70.17/86.17/66.02 74.12 | 68.07/20.05/16.62 34.91 | 24.76/36.05/27.25 29.35 | 51.85 |
| | CORAL | 65.36/96.21/45.41 68.99 | 70.20/86.15/66.01 74.12 | 68.03/20.04/16.62 34.9 | 24.70/36.05/27.24 29.33 | 51.84 |
| | SagNet | 66.60/93.41/55.61 71.87 | 61.42/79.76/64.93 68.7 | 69.04/30.38/25.88 41.77 | 26.17/36.86/25.93 29.65 | 53.0 |
| | ITL-Net | 66.30/94.67/57.29 72.75 | 74.85/86.52/75.06 78.81 | 62.60/45.82/51.44 53.29 | 17.87/26.45/19.64 21.32 | **56.54** |
| | Source set | L100 | L38 | L43 | L46 | Avg. |
| TerraIncognita | ERM | 43.82/60.93/69.15 57.97 | 39.97/51.27/54.40 48.55 | 40.96/39.11/64.70 48.26 | 58.26/46.53/73.73 59.51 | 53.57 |
| | CORAL | 43.80/60.90/69.17 57.96 | 40.00/51.12/54.20 48.44 | 40.87/40.12/64.23 48.41 | 58.70/45.43/73.62 59.25 | 53.51 |
| | SagNet | 40.71/56.95/68.19 55.28 | 37.95/50.44/53.71 47.37 | 33.99/34.41/59.01 42.47 | 59.54/45.93/74.72 60.06 | 51.3 |
| | ITL-Net | 44.79/55.71/67.62 56.04 | 44.66/53.62/58.07 52.12 | 40.97/39.29/66.35 48.87 | 61.68/51.36/76.41 63.15 | **55.04** |
| | Source set | Artistic | Clipart | Product | Real World | Avg. |
| OfficeHome | ERM | 44.75/23.36/21.09 29.73 | 39.50/18.66/20.11 26.09 | 37.45/26.00/39.55 34.33 | 27.67/31.44/56.00 38.37 | 32.13 |
| | CORAL | 44.75/23.36/21.09 29.73 | 39.49/18.67/20.11 26.09 | 37.45/26.01/39.55 34.34 | 27.77/31.46/55.98 38.4 | 32.14 |
| | SagNet | 47.98/22.33/17.85 29.39 | 46.40/23.98/14.89 28.42 | 34.84/34.34/41.62 36.93 | 33.30/35.41/54.25 40.99 | 33.93 |
| | ITL-Net | 31.63/21.57/21.53 24.91 | 54.72/21.79/30.99 35.83 | 37.38/38.75/36.69 37.61 | 33.71/35.64/56.64 42.0 | **35.09** |

A.3 THE LEARNED LOSS FUNCTION

Recall that our experimental design trained a single loss function on RotatedMNIST, and then evaluated it from different perspectives across all our main experiments. The specific parameters of our learned ITL (as visualised in Fig. 5(right)) are given in Table A.3. Then reader can plug-and-play for their own Domain Generalisation problems.

Table 8: The parameters of the learned ITL

| Loss type | parameters $\omega_0, \omega_2, ..., \omega_{11}$ |
|---|---|
| ITL | -2.0193, -1.2234, 0.1363, 0.1269, -0.4566, -0.1016, -0.2545, 1.0971, -0.9203, 0.2368, 0.4795, 0.9975 |

A.4 THE PARAMETERISATION OF THE LEARNABLE LOSS FUNCTION

We apply fourth order bi-variate Taylor polynomial to parameterise the learnable loss function. The terms only contain $\mathbf{y}_i$ are removed from the polynomial since these do not generate gradients with respect to the prediction of the network. The final form, only containing 12 learnable parameters, is given as:

$$\mathcal{L}_\omega^{(i)}(\hat{\mathbf{y}}_i, \mathbf{y}_i) = \omega_2(\hat{\mathbf{y}}_i - \omega_0) + \omega_3(\hat{\mathbf{y}}_i - \omega_0)^2 + \omega_4(\hat{\mathbf{y}}_i - \omega_0)^3 + \omega_5(\hat{\mathbf{y}}_i - \omega_0)^4$$
$$+ \omega_6(\hat{\mathbf{y}}_i - \omega_0)(\mathbf{y}_i - \omega_1) + \omega_7(\hat{\mathbf{y}}_i - \omega_0)(\mathbf{y}_i - \omega_1)^2 + \omega_8(\hat{\mathbf{y}}_i - \omega_0)^2(\mathbf{y}_i - \omega_1)$$
$$+ \omega_9(\hat{\mathbf{y}}_i - \omega_0)^3(\mathbf{y}_i - \omega_1) + \omega_{10}(\hat{\mathbf{y}}_i - \omega_0)(\mathbf{y}_i - \omega_1)^3 + \omega_{11}(\hat{\mathbf{y}}_i - \omega_0)^2(\mathbf{y}_i - \omega_1)^2.$$

A.5 ROBUST LOSS FUNCITON ILLUSTRATION

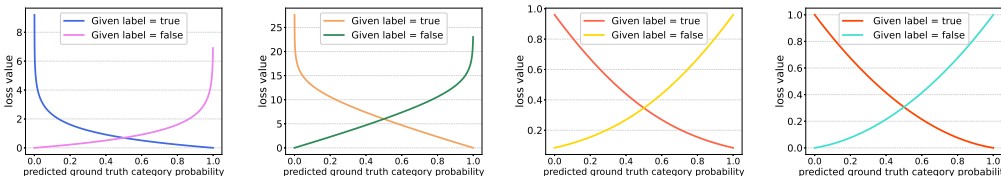

Figure 5: Comparison of loss functions (from left to right): CE, SCE, FOCAL and ITL. The range of ITL is normalised between 0 and 1.

A.6 META-TRAIN COMPUTE COST

Due to the efficiency of implicit gradient, training our ITL required using PyTorch (Paszke et al., 2017) only required 8 hours on a single V100 GPU to complete 200 gradient descent steps on $\omega$. While the goals and base learning problems are not directly comparable, this is dramatically faster than alternatives that require an entire cluster (Li et al., 2019a), and where even very recent fast methods require about 12 GPU-days (Liu et al., 2021).

A.7 META-TRAINING CONVERGENCE

Figure 6 shows the convergence of ITL during meta-training on R-MNIST. The x-axis shows outer loop iterations/loss function updates. The lines graph (i) the inner loop accuracy (mean over all source domains) after model training with the current loss function, and (ii) and the outer loop accuracy (held out domain accuracy). The convergence process is quite smooth.

A.8 FULL LOSS LANDSCAPE PLOTS

Figure 7 shows the landscape for all four domains, of which two were shown in Figure 4 of the main manuscript.

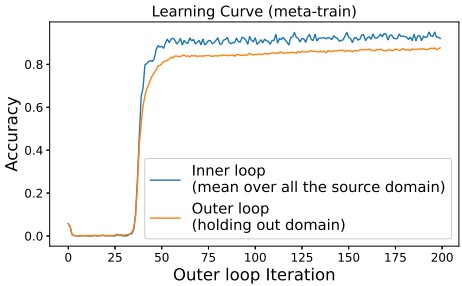

Figure 6: The learning curve for ITL meta-training stage.

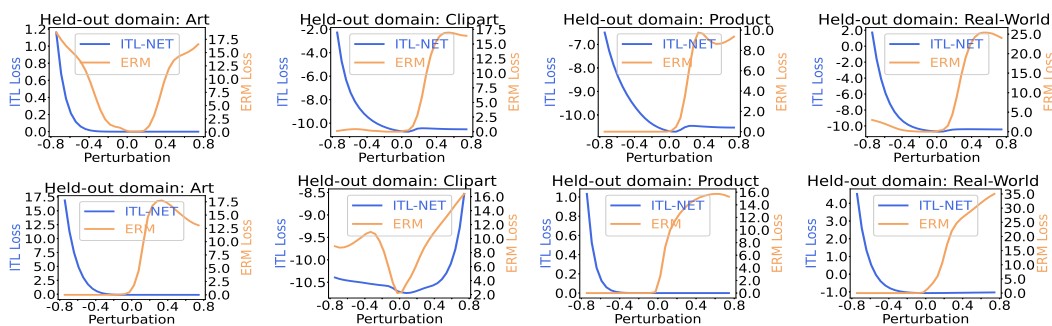

Figure 7: 1D Loss Landscape: ITL-Net vs ERM on OfficeHome. Top row: Source domain loss landscape. Bottom row: Target domain loss landscape.

