# OpenReview forum: "Loss Function Learning for Domain Generalization by Implicit Gradient"
_ICLR.cc/2022/Conference — ICLR 2022 Submitted_

### Official Review · Reviewer_Hx46 · 2021-10-19

**Correctness:** 4
**Technical Novelty And Significance:** 3
**Empirical Novelty And Significance:** 4
**Recommendation:** 8
**Confidence:** 3

**Main Review:**

Strengths:

+ The paper is well written and easy to follow, the idea is presented clearly.

+ The idea is novel and interesting; the empirical observation that learned losses transfer to new tasks results in a practical framework.

+ An extensive empirical evaluation is carried out, comparing the proposal against a diverse set of alternative methods and using a number of benchmarks, clearly supporting authors' claims.

+ The proposed loss can be directly combined with other mechanisms to further improve performance; e.g., it could be used along with some invariance-inducing strategy.


Weaknesses/questions/suggestions:

- It's possible that the proposal requires assumptions that are not discussed in the manuscript. In particular, empirical evaluation is carried out on the more common cases where data marginals shift, but data-conditional label distributions P(y|x) are fixed across domains; i.e., observing x suffices in order to determine y. It's unclear whether the learned losses would still yield improvements in the more general cases where P(y|x) is domain-dependent, i.e. the same x could have different labels depending on the domain it was observed from; C.f. [1] for an in-depth discussion on the effects of conditional shifts. In particular, if the proposal induces some type of domain-invariance, be it in the feature or prediction level, it would likely damage performance relative to standard ERM in such cases.

- The evaluation/discussion lacks in determining how learned losses improve models' robustness; e.g., are resulting models more domain-invariant in some sense? If not, what is it that makes them more robust than models training with standard ERM?

- The fact that learned losses transfer to new tasks is clearly supported by the reported evaluation. However, it would be interesting to get an idea as to whether one could benefit from training a loss tailored to a particular set of domains. In that case, it would be also interesting to get an idea of the computational cost involved in running Algorithm 1 in a larger-scale task such as PACS.

- In section 3, the text makes it a bit confusing to determine what are the authors' original contributions and what are the bits that were introduced previously and re-used in the proposal.

- Please include in the manuscript a brief description of the "gradient surgery" scheme used to aggregate gradients in Algorithm 1. Is there any reason why this particular approach would work better than simply averaging?

- Authors follow the common practice within domain generalization literature and perform evaluations using models pre-trained on ImageNet. However, in my opinion, doing so renders comparisons noisy and it's unclear how much pre-training helps in generalization to new domains. As a suggestion, I recommend repeating a subset of the evaluations without pre-training and determining whether conclusions hold.

- What is the cross-validation criterion used for experiments on PACS?

- Please report the sample size used for the significance test reported in Table 2.

References:

[1] Zhao, Han, et al. "On learning invariant representations for domain adaptation." International Conference on Machine Learning. PMLR, 2019.

**Summary Of The Paper:**

Authors introduce a bi-level optimization procedure aiming to learn parametric loss functions that result in predictors able to better generalize to new data sources, unseen during training. Specifically, an outer optimization loop iteratively updates a parametric loss function to minimize the empirical risk (under a standard loss) on a left-out data source, not presented to the model during the inner optimization loop, which trains the model on a set of domains using the current version of the parametric loss. The proposed approach is computationally expensive, but authors empirically showed that losses learned in low-scale tasks directly transfer to other cases, rendering the proposal practical.

**Summary Of The Review:**

The proposal is novel to my knowledge and interesting. It can also be directly combined with other approaches to further improve performance. The execution is robust and covers a number of well-known datasets; comparisons include recent alternative methods. Results include significance tests.

---

> ### Author Response · Authors · 2021-11-18
> **Response to reviewer**
>
> **Q1: Assumption on type of domain shift?**
> 1. Thanks for this interesting point. Actually, we don’t make any particular assumption about this, as with the case of vanilla ERM+CE. We are just changing the objective from ERM+CE to ERM+ITL.
> 2. All the benchmarks we used have p(x) shift. So it is fair to say that while we don’t make a theoretical assumption in this regard, our empirical evaluation confirmation is currently restricted to marginal shift only (as per the most popular DG evaluation benchmarks which we have used). Confirming whether changing the loss from EMR+CE to EMR+ITL holds in case of conditional p(y|x) shift remains for future work.
>
>
> **Q2: Explanation for good ITL performance?** \
> We analysed potential reasons why models trained by ITL have better performance than those trained by Cross-Entropy from two main perspectives, the prediction entropy on the target domain and landscape of the loss function.
> 1. In Figure 2, it can be noticed that model trained by ITL has relatively low entropy (low confidence) predictions, which may be due to ITL giving less aggressive feedback for wrong predictions compared to CE, and thus not forcing the model to make high confidences predictions. Unlike ITL, CE has the opposite behaviour which causes overfitting in general and in the domain generalisation case, it makes the model overfit to the source domains. Focal loss and Label smooth behave similarly to ITL, so on the single source domain generalisation, these two losses perform well.
> 2. Following the recent empirical studies  [A,B] analysing model generalisation from a loss landscape perspective, the loss landscape at convergence using ITL is flatter than that obtained when using CE, which is another reason the model trained by ITL has better performance (Fig 4)
> 3. Yet another perspective is perturbation robustness as shown in Fig 3.
>
> **Q3: Task-specific loss function?** \
> Thanks for the suggestion. Our proposed method is indeed potentially capable of learning a task-specific loss function, and this may outperform a completely general purpose loss function. However, this is expensive when using a larger task in the inner loop (currently we used R-MNIST). For example, PACS takes about 30 minutes per held-out domain, so 200 outer iterations x 2 hours = 400 GPU-hours, even assuming it takes the same number of outer iterations to converge. So we could not finish this experiment in time. We will investigate this in future work.
>
>
> **Q4: Section 3: Clarifying contributions.** \
> The contribution of the paper is divided into two-fold. First, we develop a loss function learning algorithm to produce the loss function and, secondly, we analyse the learned loss function by deploying it to different DG tasks such as DomainBed.
>
> **Q5: Gradient surgery** \
> The high-level idea of gradient surgery in the original paper is that it reduces the gradient noise which is caused by the different gradient directions generated from different tasks in multi-task learning. In our case, it will reduce the implicit gradient noise from different domain generalisation tasks in the meta-train stage, which leads to faster convergence of the loss function learning. We add a small description in the revised submission.
>
> **Q6: Domain Generalisation without pre-trained model** \
> To our knowledge existing DG studies always use pre-trained models to our knowledge. So we follow this setting to enable direct comparison to prior work.
> We agree this is an interesting question, but since it applies generally to all DG methods and not only ours, it is getting out of the scope of this paper, and we leave it to future work.
>
> **Q7: Cross-validation criterion used on PACS** \
> On PACS, we use the validation set of the source domain for cross-validation in Table 1, following the splitting and validation protocol defined in the original PACS dataset release. For Table 2, we still use the source domain validation set but follow  DomainBed dataset splitting protocol as implemented by DomainBed framework.
>
> **Q8: Sample size for the hypothesis testing** \
> The sample size used for the hypothesis testing equals the number of datasets we used in DomainBed which is 6. We made it clear in the latest revised submission.
>
>
> **Reference** \
> [A] Keskar, Nitish Shirish, et al. "On Large-Batch Training for Deep Learning: Generalization Gap and Sharp Minima." in ICLR, 2017. \
> [B] Chaudhari, Pratik, et al. "Entropy-SGD: biasing gradient descent into wide valleys." Journal of Statistical Mechanics: Theory and Experiment 12.12 (2019): 124018.

---

### Official Review · Reviewer_eZ71 · 2021-11-02

**Correctness:** 2
**Technical Novelty And Significance:** 3
**Empirical Novelty And Significance:** 3
**Recommendation:** 3
**Confidence:** 4

**Details Of Ethics Concerns:**

The authors are encouraged to include a statement of their potential fairness issues. For example, the observed source environments can be biased for the test domain.

-- Final update after discussion

The author still does not provide an ethical statement.


**Main Review:**

Overall the paper is interesting since the author provides an alternative perspective in domain generalization: searching the proper loss function.  However, this reviewer still found a couple of important concerns (including theory, method, and paper writing) that prevent me from accepting the current version. Based on these, I believe a major revision and rethinking about the cons can significantly improve the paper.

Pros:

- This paper proposes a novel direction in domain generalization: searching the proper loss. Although this idea is not completely novel in the related domains such as NAS or AutoML, applying them to the domain generalization seems quite interesting and novel.
- The empirical studies and analysis are extensive.

Cons (see detailed reviews for explanations):

- The main concern lies in the theoretical aspects: why and when it works or fails? The proposed idea currently seems a bit ad-hoc.
- The paper is not self-contained. Some details are quite difficult to follow or lack motivation.

---------------Detailed reviews

**About theoretical understanding.**

I totally agree with the author’s viewpoint about the problem revealed in domainBed. However, in this paper, the author fails to clearly justify why the proposed parametric loss $w$ can effectively solve this problem. Specifically,

1. Does the bi-level optimization indeed search for the correct parametric loss rather than over-fitting? Note in the domain generalization, the source number is generally quite limited (different from NAS), supposing you only have 3 sources, it is quite difficult to show the bi-level indeed finding the best configuration. (It is more likely overfitting). To address this, the author should provide a clear theoretical analysis to show when the proposed method can work.

2. The selected Taylor Polynomial representation is wired and lacks motivation. Why do we choose this specific loss? What is the convergence behavior in the bi-level optimization if this loss is adopted? Why not MSE loss? Triple loss? This paper lacks a clear motivation in illustrating why it is essential.

3. The experimental settings in the single DG are also wired. Clearly, one source could not generalize to other related targets. Therefore I am a bit doubtful about the proposed method...

**Other technique details**

- Some parts are not self-contained and unclear. For example,

1. Algo 1, Line 15 there is no description of the grad-surgery approach.

2. Algo1, Line 16. h{update...} what does it mean?

3. In algorithm 2, the description seems inconsistent with the sum of the Neumann series. The role of j is missing in the loop.

4. Computing jacobian-vector product in PyTorch seems a bit difficult if we directly adopt PyTorch since PyTorch currently does not support the form (algorithm 2) Jacobian-vector-product(parameter of neural-network,\theta,v). I am wondering how you addressed this.

- The convergence bound of using Neumann series or Implicit gradient is highly expected to provide. (I think it is feasible under proper assumptions.)

- There may exist a memory complex concern since the $H$ in the algorithm depends on the task number. If n is large, the memory complexity can be quite high (although the implicit function addressed the computational complexity).









**Summary Of The Paper:**

This paper proposed a novel viewpoint in domain generalization, i.e, the loss function search. Specifically, the search procedure is decomposed as a bi-level optimization and solved through implicit function. The author later adopted the Neumann series for memory saving. The extensive empirical results validate the correctness of the proposed approach.

====== Final Update after rolling discussions

I would appreciate the authors for the detailed responses and I have read the rebuttal. Unfortunately, the concern of rigor is still not addressed after discussion and I will maintain the same rating. Below are my *final* remarks (I fully understand your high-level idea), I hope the author will carefully rethink the paper and improve it in the future version.

- In the rebuttal and paper, many terms are not mathematically/rigorously defined, which made me rather confused. These terms even made me doubtful about the generalization properties since they should be rigorously defined and discussed. (see Note 1 for the detailed examples)

- The motivation for choosing the specific loss parameter families is not properly addressed in the paper. From the rebuttal, I finally got the logic/motivation but it was too late. I do think a major revision for fully justifying the motivation can greatly improve the paper. (see Note 2 for the detailed discussions)

---------------------

Note 1.  Examples of unclear terms, most of them are not formally formulated/defined. The descriptive words rather than the math formula made the reviewer confused and a bit doubtful about the generalization property since generalization should be rigorously defined.

  - "Scale of the domain shift is similar enough between meta-train and deployment datasets." What is the formal definition of the scale of domain shift?

 -  “the cardinality of the classification problem is not too different between meta-train and meta-test.” What is the formal definition of the cardinality of classification?

  - “But please note that something like MNIST->SVHN transfer is not exactly the problem setting we address.” It is not domain generalization (trained on digit A and deployed on digits B)? What is the formal (or mathematic) definition of a single source setting? It is best to use equations rather than descriptive words to depict the problem settings. In fact, the single source is a confusing word (since it is not mathematically defined.)

 -  “large search space is expected to lead to meta-overfitting.” What is the formal definition of meta-overfitting? Overfitting w.r.t. which distribution? The author should be cautious to use unclear words since meta-overfitting has not been mathematically defined.


Note 2. The motivation of choosing the specific loss parameter.

Thanks for the rebuttal. It makes me clear about the motivation. But it requires a major revision of your paper to fully justify it. For example, I would suggest testing the linear combination and showing its results. Also exploring other loss families to fully justify the value of the loss search. In the current version, it is more or less we proposed loss family A (without strong motivation or benefits) and it works in some classification benchmark.  A *comprehensive and systematic* empirical understanding can significantly improve it.

-------------------------------------------------------


**Summary Of The Review:**

This paper adopts the idea of NAS or autoML to search for the proper loss in the domain generalization. Overall the idea sounds novel in the domain generalization while the current versions lack clear justifications (in terms of theory). Based on these, I do not recommend accepting the current version but encourage a major revision for the resubmission.

---

> ### Author Response · Authors · 2021-11-18
> **Response to reviewer**
>
> **Q1: Does the bi-level optimization indeed search for the correct parametric loss rather than over-fitting?**
> 1. While we don’t have theoretical proof. Our experiments clearly demonstrate that the loss does indeed generalise across experiments (as also acknowledged by Reviewer Hx46). There is no overlap between meta-train/meta-test datasets, so our results show that the loss does not overfit. Our loss is learned on RotatedMNIST and then applied to all 6 datasets in DomainBed benchmark and generalises across all. We add an algorithm diagram to show the idea of learning and deploying the loss pipeline.
> 2. Standard losses such as CE/MSE/etc losses generalise across problems. Our loss is a small polynomial expression analogous to MSE, etc (actually it includes MSE as a special case). Therefore it should not be surprising that it can generalise across problems.
> 3. We specifically chose polynomials as a family of loss functions with few parameters (see also the response to Q2). Since our loss only has *twelve* parameters, it is less likely to overfit to a particular domain compared to more common, e.g., neural losses [B,C].
>
> **Q2: Taylor Polynomial representation**
> 1. Our loss family is not weird. Other studies have successfully demonstrated that polynomial losses can generalise across different problems [A].
> 2. We plot the inner and outer loop learning curve to show the convergence behaviour of the proposed algorithm in Figure 6/appendix A.7 in the latest revised submission.
> 3. Why not MSE/etc? Vanilla MSE or Triplet loss does not have a learnable parametric form, so there is nothing to learn. Note that MSE is a special case of our Taylor polynomial. So our search space includes the possibility of discovering MSE.
> 4. To recap: (i) We need a loss function with some learnable parameters in order to have a chance to improve on hand-chosen losses (MSE, CE, etc). (ii) We need a loss function without so many learnable parameters that it overfits to the source data. This is why we chose a polynomial family loss [A] with 12 parameters, rather than neural family losses like [B,C] which can have 10,000s of parameters.
>
>
> **Q3: Single source domain DG** \
> We respectfully disagree with the assertion “clearly one source could not generalise”. Cross-task generalisation of learned parameters is widely achieved in fields such as NAS. In the case of loss learning, actually, several papers [A,C,D,E] learned a loss on dataset X and successfully deployed it on dataset Y. So there is precedent for this and it is definitely not obviously impossible as claimed. Our contribution is to show that such transferable losses can be learned with the objective of improving DG performance, rather than supervised learning performance as in [A,D,E].
>
> **Other technique details**
> 1. The reference of grad-surgery is given in the last line of section 3.3.
> 2. Gradient surgery generates the update information, gradient h, computed from a group of implicit gradients which is computed based on H. {update...} is an algorithm comment explaining the line: Parameters of the loss function are updated by the gradient, h.
> 3. j is described in Eq 6.
> 4. We can do such an operation in Pytorch. Algorithm 2 already gives the implementation of JVP by using torch.autograd.grad and in the recent version Pytorch provides torch.autograd.functional.jvp API. **Link:** https://pytorch.org/docs/stable/generated/torch.autograd.functional.jvp.html
> 5. H is n \times 12 where 12 is the number of parameters in the learnable loss function. It will not be very expensive. We further reduce the computation complexity by learning the loss function on the toy domain generalisation dataset, rotated MNIST, where only small neural networks are needed. In addition, n is 5 when using rotated MNIST for meta-train.
>
>
>
> **Reference** \
> [A] Gonzalez, Santiago, and Risto Miikkulainen. "Optimizing loss functions through multi-variate Taylor polynomial parameterization." Proceedings of the Genetic and Evolutionary Computation Conference. 2021. \
> [B] Li, Yiying, et al. "Feature-critic networks for heterogeneous domain generalization." International Conference on Machine Learning. PMLR, 2019. \
> [C] Bechtle, Sarah, et al. "Meta learning via learned loss." 2020 25th International Conference on Pattern Recognition (ICPR). IEEE, 2021. \
> [D] Tao, Chenxin et al. "Searching Parameterized AP Loss for Object Detection." Thirty-Fifth Conference on Nueral Information Processing Systems (NeurIPS), 2021. \
> [E] Li, Hao, et al. "Auto Seg-Loss: Searching Metric Surrogates for Semantic Segmentation." International Conference on Learning Representations. 2020. \
> [F] Li, Da, et al. "Deeper, broader and artier domain generalization." Proceedings of the IEEE international conference on computer vision. 2017.

---

> ### Comment · Reviewer_eZ71 · 2021-11-24
> **Post rebuttal**
>
> Dear author,
>
> I appreciate your detailed responses. Unfortunately, my main concern “a lack of rigor” is still not addressed, rather the rebuttal enforces the impression that the idea is ad-hoc and lacks rigor.  I hope the additional comments can help the authors rethink, provide **rigorous** justifications and illustrate the limitations of the proposed approach. Otherwise, the proposed idea and research can be misleading for the community.
>
> 1. The paper and rebuttal claimed multiple times the empirical results clearly outperforms by changing the parameter of the loss. I am still not convinced this discovery can be applied to a general setting when it is out of the benchmark. For example, we consider the source is only MNIST, the test dataset is SVHN, color-Mnist, it is highly probable the learned parametric loss could not have a good performance on such dataset, although they are semantically quite similar (digits recognition). Based on this, the author should clearly discuss the *assumption* when the proposed approach works or fails. (This does not require that the author should prove some formal results, but rather rigorously justify and discuss its limitations.)
>
> 2. Again, the proposed parametric loss still lacks a clear motivation of why choosing such a family. What are the **specific** benefits? Otherwise, I could adopt multiple classification losses and use the linear combination for such different losses. Thus a meta-learning procedure is to learn the combination coefficient. Why not this idea? If I fine-tune carefully, the accuracy can be better than ERM.
>
> 3. Hyper-parameter optimization is quite different from domain generalization. In fact, the validation set and training set are different empirical distributions but generally, we assume they come from the same underlying distribution. (e.g, training and validation are all MNIST, validation will not be SVHN.) In domain generalization, what are the corresponding assumptions? Which assumption could you conduct the meta-train and meta-test? How to justify it?
>
> 4. About single source, See the example in Q1. Is it possible the proposed approach trained on MNIST and works for SVHN, although they are all digits recognition?
>
>
> Other comments
> - I am still unclear about the gradient surgery approach. The author did not update a clear algorithm in their revised version for those who are unfamiliar with this notion.
>
> - > We can do such an operation in Pytorch. Algorithm 2 already gives the implementation of JVP by using torch.autograd.grad and in the recent version Pytorch provides torch.autograd.functional.jvp API.
>
> My question still remains. I would like to see how you use torch.autograd.grad to compute this. The torch.autograd.functional.jvp generally does not support the input as the parameter of the neural network.
>
> - >  H is n \times 12 where 12 is the number of parameters in the learnable loss function.We further reduce the computation complexity by learning the loss function on the toy domain generalisation dataset, rotated MNIST, where only small neural networks are needed. In addition, n is 5 when using rotated MNIST for meta-train.
>
> Again this statement exactly suggests the proposed approach lacks rigor. If we change the parametric loss with a large search space, this idea does not scale.
>
> Overall, it is highly expected the author carefully rethink the rigor in their proposed approach. We should be very cautious when we talk about **generalization** since an improper discussion could mislead the community.

---

> > ### Author Response · Authors · 2021-11-29
> > **Response to reviewer**
> >
> > ***1. Can we expect losses to generalise across datasets such as MNIST -> SVHN? Hyperparameter optimization is different because train, val and test are from the same distribution? What are the assumptions?*** \
> > Precedent: Please let us first reiterate that there *is* precedent for meta-learning generalisation *across domains*. To give a few examples: In DARTS [A] they search for a neural architecture on CIFAR-10, and then deploy the discovered architecture for successful learning on ImageNet. In metaperturb [B] they train a dropout regularizer on TinyImageNet, and deploy it to learn a suite of tasks (CIFAR-100, Stanford Dogs, Stanford Cards, Aircraft, CUB). In GLO [C] they searched for a loss function on MNIST, and then deployed it to successfully learn on CIFAR-10 with better performance than regular cross-entropy - even though MNIST and CIFAR-10 have neither semantic nor low-level statistical similarity. So we hope the reviewer can agree that it is not necessarily impossible to transfer meta-learning knowledge across domain/dataset.
> >
> > Assumptions and Limitations: That said, we agree with the reviewer that it would be beneficial to elucidate implicit assumptions on when and why one should expect loss learning to work or fail in the domain generalisation setting, which we are the first to address here. A key difference between [A,B,C] and ours is that we are explicitly trying to explicitly learn about domain-shifts to improve robustness. First, please recall (Fig 1) that during-meta learning we use *domain-shifted (non-IID) dataset* splits for train/inner loop (Eq 2) and validation/outer loop (Eq 3). For example, different image rotations in R-MNIST. What we expect here is that because our loss is meta-learned to improve performance in the case where data is non-IID between train and val, that it will regularize more strongly than cross-entropy (which is the defacto choice for standard learning due to its good performance on IID train + val data). In contrast, if we had used IID splits from MNIST in Eq2 and Eq3, we expect the performance of the learned loss would be the same or worse than cross-entropy when deployed in a novel DG problem. Following this point, an implicit assumption is that the *degree* of domain shift among domains is similar enough between the meta-training dataset (R-MNIST domains), and meta-testing dataset (e.g., PACS domains, OfficeHome domains, etc). If this assumption is violated, the degree of regularisation learned in the loss may be stronger or weaker than is ideal for the target domain. For example, if we deployed our meta-learned ITL to a learning problem with IID train and val/test (i.e., without domain shift), then performance is expected to be worse than cross-entropy.
> > Note that this is a related but weaker generalisation of the usual assumption most of the multi-source DG algorithms: They expect the domain-shift exhibited among the source domains (e.g., P,A,C in PACS) to be similar to be representative of the domain-shift exhibited by the target domain (e.g., S in PACS). For DG algorithms they learn an entire NN with thousands of parameters, so the shifts have to be well matched to avoid overfitting to the source. For us, we only learn a dozen parameters, so the meta-training and meta-testing domain-shifts need not be very similar.
> >
> > A final limitation to mention is that while we would expect not much issue transferring between SVHN and MNIST (because our loss inputs posterior probabilities and 1-hot vector labels, the low-level image statistics are not so relevant), performance may be comparatively weaker if the cardinality is dramatically different between meta-train (10-way in MNIST) and meta-test (e.g., 1000-way in ImageNet).
> >
> > So to summarize the assumptions: (1) we need a multi-domain training set for meta-learning, (2) the scale of the domain shift is similar enough between meta-train and deployment datasets, (3) the cardinality of the classification problem is not too different between meta-train and meta-test. (4) The loss parametrisation must be (i) low dimensional loss to minimise overfitting to the meta-training dataset, (ii) smoothly parameterised to allow differentiable optimization.
> >
> > [A] Liu, ICLR’19, DARTS: Differentiable Architecture Search \
> > [B] Ryu, NeurIPS’20, MetaPerturb: Transferable Regularizer for Heterogeneous Tasks and Architectures \
> > [C] Gonzalez, CEC’20, Improved Training Speed, Accuracy, and Data Utilization Through Loss Function Optimization

---

> > ### Author Response · Authors · 2021-11-29
> > **Response to reviewer**
> >
> > ***2. Motivation for loss family? Why not linear combination?*** \
> > Thanks. First, we reiterate that we are not proposing a new loss family. Our choice of loss family is from [A]. Please see [A] for extensive discussion on why Taylor polynomials as opposed to various other possible loss parameterisations. We hope the reviewer can agree that it must be a low-dimensional parameterization. (Because if it was a high dimensional parameterisation like a neural network, it would be too easy to overfit to the specific statistics of the source-dataset label distribution.). In terms of other low dimensional parametrisations, linear combinations are a reasonable alternative. In fact, we tried that earlier in this project. A previously published loss, Symmetric Cross-Entropy [B] proposes to optimise LossSCE = CE(p||q) + CE(q||p). We used our same meta-learning pipeline to train hyper-parameters (alpha,beta) in LossMetaSCE = alpha*CE(p||q)+beta*CE(q||p). This also worked OK. It improved on vanilla CE, but it did not work as well as the taylor polynomials from [A], which we eventually focused on. Other low dimensional parameterisations such as those used in GLO [C] are not smooth and differentiable for easy optimization. Finally, note that we are not saying Taylor polynomials are the best possible loss. But they are better than many other options explored so far.
> >
> > ***3. Single source. Is it possible that [losses] trained on MNIST work for SVHN?*** \
> > Yes. Cross-dataset loss transfer was already demonstrated in [C] for MNIST->CIFAR. But please note that something like MNIST->SVHN transfer is not exactly the problem setting we address.
> > In the multi-source case, we train the loss on a *set of domains*, like all the R-MNIST rotations, and deploy the loss for learning on a different *set of domains* like Photo/Art/Cartoon/Sketch in PACS.
> > In the single source case, we train the loss on a source *set of domains* like a subset of the R-MNIST rotations, use the loss to train a model on the same source set of domains, and then deploy the trained model on a new target domain (like a held-out R-MNIST rotation).
> >
> > ***4. Algorithm and torch autograd.*** \
> > As stated in both the paper and our first response—which is quoted by the reviewer—we show in Algorithm 1 & 2 how to use torch.autograd.grad to compute the relevant hypergradient.  For completeness, we report the exact pytorch code below.
> >
> >
> > ```
> > def algorithm2(Lw, M, omega, theta, J, alpha):
> >     dLw_dtheta = torch.autograd.grad(Lw, theta, allow_unused=True, create_graph=True)
> >     v = p = torch.autograd.grad(M, theta, allow_unused= True, create_graph= True)
> >
> >     for j in range(J):
> >         grad = torch.autograd.grad(
> >             dLw_dtheta,
> >             theta,
> >             grad_outputs=v,
> >             retain_graph=True,
> >             allow_unused=True
> >         )
> >
> >         grad = [g_ele * alpha for g_ele in grad]
> >
> >         v = [v_ele - g_ele for (v_ele, g_ele) in zip(v, grad)]
> >         p = [p_ele + v_ele for (p_ele, v_ele) in zip(p, v)]
> >
> >     tmp = list(p_tmp for p_tmp in p)
> >
> >     hypergradient = torch.autograd.grad(
> >         dLw_dtheta,
> >         omega,
> >         grad_outputs=tmp,
> >         allow_unused=True
> >     )
> >
> >     return list(-g_ele for g_ele in hypergradient)
> > ```
> > ***5. Proposed approach lacks rigor. If we change the parametric loss with a large search space, this idea does not scale?*** \
> > It is unclear what the reviewer means by rigor in this context. With regard to the scalability point, we emphasize: (1) We are in any case only interested in small search spaces because using a large search space is expected to lead to meta-overfitting, in which our cross-dataset transfer might no longer succeed (see clarified assumptions in Q1). (2) There is no scalability barrier as implied by the reviewer. Assuming hypothetically that meta-overfitting was not an issue due, e.g., to a huge source domain set. In this case, there is no intrinsic computational problem to optimize a high-dimensional loss with our method. The IFT-based strategy that we use is famous for being extensible to optimizing millions of hyper-parameters [D]. So there is definitely no scalability problem.
> >
> > [A] Gonzalez, GECCO’21, Optimizing Loss Functions through Multi-Variate Taylor Polynomial Parameterization. \
> > [B] Wang, ICCV’19, Symmetric cross entropy for robust learning with noisy labels \
> > [C] Gonzalez, CEC’20, Improved Training Speed, Accuracy, and Data Utilization Through Loss Function Optimization \
> > [D] Lorraine, AISTATS’20, Optimizing millions of hyperparameters by implicit differentiation

---

### Official Review · Reviewer_57HS · 2021-11-04

**Correctness:** 3
**Technical Novelty And Significance:** 4
**Empirical Novelty And Significance:** 3
**Recommendation:** 6
**Confidence:** 3

**Main Review:**

The paper is well written and well structured. The idea of meta-learning a loss function based on a given domain generalization task is interesting and (to the best of my knowledge) novel. The experimental setup (especially on DomainBed) appears rigorous (but see my comments below).

### Weaknesses:

- The final loss function used is actually not printed in in the paper---or did I miss it? Given that the whole result section is meant to show improvements of using this loss function, it should be added to the main paper along with information about any implementational details needed for re-implementing it.
- Table 2 is missing confidence intervals or standard deviations / standard errors. Otherwise the differences (sometimes as small as .1 percentage points) are hard to judge. For example, the original paper reported 41.8% for CORAL on DomainNet, vs. 41.5% reported here, so there seems to be some variantion. As an alternative suggestion, what about showing the average rank of the method across all domains in a dataset in brackets behind the performance?
- §4.3, Repeatability analysis: ColoredMNIST seems to be a bad task choice for this claim, as all results in the paper essentially fail on the task (which is expected---it was specifically designed to show an advantage of the Invariant Risk Minimization algorithm). I suggest to re-run the experiment in Table 5 on a different dataset.
- The presentation of the results is generally misleading. At many places, section 4 reads more like an advertisement of the method, rather than a neutral assessment and fair comparision of the results. It is good that the full tables are in the appendix to give a better overview of where the method works, and where it fails. For example, on VLCS, CORAL outperforms the method on 3/4 domains, still the average performance is a bit better --- communicating the results in this form is misleading, and needs to be improved (it is simply conditional on the baseline numbers of the different domains). This is not the only example: E.g. in Table 7, the wrong number is bolded for VLCS.
- The conclusion does not match the paper story and should be improved. I highly recommend to write an extra paragraph on the limitations of the presented experiments.
- The paper lacks a good "Figure 1" that outlines the approach. I suggest one of the less interesting analysis figures (E.g. the current Figure 1) to the supplement, in case space is needed for such a figure.

### Minor points

- The plots need work: When printed, labels and legend entries are not readable.
- Tables should not have vertical lines according to the formatting instructions.
- The Table 2 caption should highlight that these results are for a ResNet-50 model.

### Additional questions

- What informed the choice of 0.025 as the signficance threshold?
- Table 2: How is it possible the the p-value for all comparisions is exactly 0.016? Could you clarify what was actually compared? Was the test performed on average performances, or by considering the ranking on individual domains? If you check all individual domains in DomainBed, what is the ranking of the considered methods? Looking at Table 6 suggestions that e.g. CORAL is often close in performance (despite being a quite old algorithm), the only clear improvements are observed on TerraIncognita and to some extend on OfficeHome.
- Abstract: "enables simple ERM to outperform signficiantly more complicated prior DG methods" --> I disagree with this statement --- the amount of work needed to come up with the loss function is arguably larger than e.g. the CORAL technique.

**Summary Of The Paper:**

The authors design a scheme for meta-learning loss functions for domain generalization: Based on the RotatedMNIST problem, a new implicit gradient algorithm is used to select a loss function from a chosen parametric family, and then applied to other domain generalization problems. The authors demonstrate improvements over the Mixstyle method from Zhou et al (ICLR'21) on PACS, and improvements over ERM and other methods on DomainBed.

**Summary Of The Review:**

The paper proposes a new interesting setting for discovering new loss functions based on a predefined task, and extensively evaluates one "proof-of-concept" loss function derived with this technique.

My main concern is that the authors occassionally oversell their results, along with some issues related to this, e.g. a potential inconsistency in the hypothesis testing done for Table 2 (see above). Realistically, it seems that the method works for a few of the settings, but not all of them---which I think is totally fine, especially in the light of the results in the DomainBed paper, that showed that a lot of methods proposed over the past years actually fail to meaningfully outperform ERM.

The authors should also critically examine their full results again, adapt the language in the paper to be more realistic about the actual merit of the method and the improvements that can be expected. This should be easy to fix by revising the language in the result and conclusion section a bit, and adding a short discussion of the failure cases.

I assign a score of (5) for now, but would be very happy to increase it --- based on the results already in the paper, I think the paper could become an interesting contribution to ICLR, if presented correctly.

---

**Post-Rebuttal comments:**

The authors sufficiently addressed my concerns, and I decided to increase my score to (6), as I think that this work might be an interesting contribution in the space of domain generalization, first and foremost for putting forward the idea of learning loss functions based on an auxilary domain generalization task itself, which to the best of my knowledge this paper provides the first proof-of-concept for.

For the camera ready, I would however urge the authors to carefully re-check the paper again, and tune down claims that are not sufficiently supported by the presented data. I would also consider to remove the statistical analysis (but keep the average ranks--these are informative), cf. our ongoing discussion, I think in the current form it does not add much information to the paper. If the authors include it, the performed test should be well justified in the supplement, which is currently not the case. I would also recommend to run a test applicable to multiple comparisons, and then perform a suitable post-hoc test, instead of the current practice of testing all models with a Wilcoxon test. It should be highlighted that especially for CORAL/SagNet vs. the proposed method, the gains are rather small in some cases.

If the authors add a bit more critical discussion on their method to the paper, I would hence recommend to accept the paper.

---

> ### Author Response · Authors · 2021-11-18
> **Response to reviewer**
>
> **Q1: Final loss function** \
> The final loss function is given in Appendix A.3 and its plot is in Figure 2 in the original submission and in Figure 5 in the revised submission.
>
> **Q2: Missing confidence interval**\
> Table 2 is the average performance over all the hold out domains, basically average over average. We follow the exact reporting protocol in DomainBed for evaluation which also does not contain std for the average performance over domains (see Tab 3 in Domain Bed paper). Please note that we do already report the mean and std for within domain results in Table 6 in Appendix A.1. The average rank over the datasets is added in the revised Table2.
>
> With regard to CORAL performance on DomainNet. This discrepancy is due to two versions of their paper, which report 41.8% and 41.5% in the Arxiv [A] and the ICLR 2021 [B] versions respectively. We choose to use the official number in the ICLR 2021 published version.
>
> **Q3: Repeatability analysis** \
> Thanks for the suggestion. We updated Table 5 by running experiments on OfficeHome which can be found in the revised submission. Either R-MNIST or R-KMNIST sources leads to discovering a loss function that improves performance on downstream OfficeHome.
>
> **Q4: Result representing problem** \
> On VLCS, Coral does have very close performance with ITL-NET. That is the reason why we conduct hypothesis testing to show that there is a significant improvement between ITL-NET and other models from the rank perspective (hence signed rank test). In the revised submission, now report the average rank for each model across datasets in Table 2.
>
> The highlighting problem about VLCS in Table 6 is solved in the revised version.
>
> **Q5: Limits of the Proposed method** \
> In the revised submission, we add the following paragraph at the end of section 4.
> ITL improves ERM+CE for DG tasks in general, but in some cases, the margin over SOTA is small, since other state-of-the-art competitors may beat ERM+CE. It would be more interesting if ITL is highly complementary to SOTA methods based on other architectural, augmentation, or domain-alignment improvements -- but this remains for future work to determine.
>
> **Q6: Overview figure** \
> We now plot a diagram to illustrate the idea of the proposed algorithm including learning and deploying the loss function to replace the old Figure 1 which has been moved on the appendix.
>
> **Minor Points** \
> The font size in the figures is increased and the highlight problem in the table captions has been solved. If you have a further concern about the format of the paper, please let us know, we will fix it.
>
> **Additional questions**
> 1. We choose confidence level as 0.025 since it is a standard operation for one side hypothesis testing.
> 2. The hypothesis test, the Wilcoxon signed-rank test, is based on the rank of each model in terms of average performance on held out domains within each dataset.  We hope the revised Tab 2 clarifies this. Since ITL has a consistent rank=1, the p-value is the same for each comparison.
> 3. We have toned this down.
>
> **Rreference** \
> [A] Arxiv DomainBed: https://arxiv.org/pdf/2007.01434.pdf \
> [B] ICLR 2021: https://openreview.net/pdf?id=lQdXeXDoWtI

---

> > ### Comment · Reviewer_57HS · 2021-11-18
> > **Re: Average rank (Table 2)**
> >
> > Thanks for the response. A short follow-up regarding the rank you now report in Table 2: In Table 6, across each invididual domains, other methods outperform ITL-Net, e.g. on V-Pascal, ITL-Net comes in 5th.
> >
> > You would paint a more realistic picture, and also strengthen the comparison by not averaging out these differences. Hence, I would still find it more appropriate to calculate the average rank (and also perform the rank test) by treating each domain individually, vs. calculating this on averaged numbers for each dataset.
> >
> > Alternatively, could you explain the rationale why testing on average numbers per dataset gives a more realistic impression on the algorithm performance?

---

> > > ### Author Response · Authors · 2021-11-23
> > > **Response to reviewer**
> > >
> > >
> > > **Why Average Accuracy** \
> > > We want to compare performance averaged over multiple unseen domains because the idea is that this is an estimator for the expected performance on held-out domains. This is the idealised objective, as introduced in [B, C], and also the performance metric that they report for the real-world applications they consider. This is the standard evaluation metric for DG. For example, [A] does all the comparisons with such evaluation. So do all the baselines we use in our paper.
> > >
> > > [A] Gulrajani, Ishaan, and David Lopez-Paz. "In Search of Lost Domain Generalization." International Conference on Learning Representations. 2020. \
> > > [B] Blanchard, Gilles, Gyemin Lee, and Clayton Scott. "Generalizing from several related classification tasks to a new unlabeled sample." Advances in neural information processing systems 24, 2011. \
> > > [C] Muandet, Krikamol, David Balduzzi, and Bernhard Schölkopf. "Domain generalization via invariant feature representation." International Conference on Machine Learning. PMLR, 2013.

---

> > > > ### Comment · Reviewer_57HS · 2021-11-28
> > > > **Re: Re: Average rank (Table 2) / Statistics**
> > > >
> > > > Thanks for the response, and apologies for the late reply.
> > > >
> > > > I really like this paragraph, thanks for including:
> > > >
> > > > > Limitations. ITL improves ERM+CE for DG tasks in general, but in some cases, the margin over SOTA is small, since other state-of-the-art competitors may beat ERM+CE. It would be more interesting if ITL is highly complementary to SOTA methods based on other architectural, augmentation, or domain-alignment improvements [...]
> > > >
> > > > That being said, I have some final concerns about the presented data, and the statistics used for analysing them:
> > > >
> > > > ---
> > > >
> > > > > The hypothesis test, the Wilcoxon signed-rank test, is based on the rank of each model in terms of average performance on held out domains within each dataset. We hope the revised Tab 2 clarifies this. Since ITL has a consistent rank=1, the p-value is the same for each comparison.
> > > >
> > > > To re-iterate my question, could you let me know why you performed the test on the average values, rather than the individual samples? I understand that other papers also show average values, but that's what you do anyways in Table 2 --- computing the average rank based on these numbers only is not an interesting addition. Providing the average rank across 24 individiual domains is -- it gives the reader a realistic picture of the performance of each method *beyond* what is already shown in Table 2.
> > > >
> > > > Same goes for the test: The test is now performed on only five numbers, vs. 24 domains you are testing. Just to put your numbers into perspective, when I re-calculate the average rank based on individual domain results from Table 6, I get the following average ranks:
> > > >
> > > > ```
> > > > ERM        3.64
> > > > SagNet     2.84
> > > > CORAL      2.40
> > > > CDANN      5.00
> > > > RSC        4.60
> > > > ITL-Net    2.04
> > > > ```
> > > >
> > > > and the difference to e.g. CORAL gets smaller, while the difference to ERM is still considerable (i.e., the result is totally fine).
> > > >
> > > > When performing a one-sided Wilconox test on all 24 numbers, I get the following p-values when comparing against the null that the depicted method in each row performs better or equal against ITL (i.e., all are significant at 5% level):
> > > >
> > > > ```
> > > > ERM: p=0.00062***
> > > > SagNet: p=0.00369**
> > > > CORAL: p=0.03163*
> > > > CDANN: p=0.00002***
> > > > RSC: p=0.00002***
> > > > ```
> > > >
> > > > This would paint a much more realistic picture than the numbers you currently give in the table, which the reader can already infer from the numbers in the table itself. I do not fully understand the motivation for performing the test in such a setting in the first place, it does not add additional information to the main paper.
> > > >
> > > > What I should note is, that the performance over CORAL is really minor. When excluding the data on Colored MNIST which in my opinion is an appropriate choice, since it is clear from the data that all algorithms simply fail (the third domain is below chance level), the rank test assigns the following p values:
> > > >
> > > > ```
> > > > ERM: p=0.00125**
> > > > SagNet: p=0.00516**
> > > > CORAL: p=0.05116 (ns)
> > > > CDANN: p=0.00003***
> > > > RSC: p=0.00003***
> > > > ```
> > > >
> > > > And this might be something worth discussing: While ERM is still outperformed, the improvement over CORAL is marginal (as I suspected in my initial comment). So it is well conceivable that in some other setups, CORAL is also a technique worth testing.
> > > >
> > > > >  To formally compare ITL-Net with competitors, we perform significance testing using the Wilcoxon signed-rank test, where the p-value is set as 0.025 and the sample size is number of the Domain datasets applied. For example, when comparing the performance of ITL-Net and ERM, the null hypothesis (H0) is that ITL-Net has equal performance to ERM on DomainBed and *the alternative hypothesis (H1) that ITL-Net is statistically significantly better than ERM on DomainBed*.
> > > >
> > > > I might miss something, but from what you wrote so far, this is wrong if you performed a two-sided test: For a two-sided test, the null hypothesis is that both algorithms are equal. Why do you use a two-sided over a one-sided test, shouldn't your null hypothesis include the option that ITL performs worse than the other methods? Even if you leave it at the two-sided test, the sentence needs to be updated. Please let me know how you plan to address this.
> > > >
> > > > **Data inconsistencies**
> > > >
> > > > While looking at Table 6, I found the following inconsistencies in the data (please double check these and also update Table 2 where needed):
> > > >
> > > > * VLCS, ERM: Avg is 78.2%, not 77.5%
> > > > * OfficeHome, CDANN, Avg is 65.6%, not 65.8%
> > > >
> > > > The last one could be due to rounding -- For the first one, could you let me know how these made it into the paper, and whether I can expect to find additional ones in other tables? I would suggest to carefully check the values in the tables again, and whether this changes any statistics.
> > > >
> > > > Also in Table 7, could you please confirm that it is correct that on VLCS, the numbers for ERM and CORAL are *exactly* the same?
> > > >
> > > > ---
> > > >
> > > > I would appreciate some final thoughts on this topic, and apologize for the late reply.

---

> > > > > ### Author Response · Authors · 2021-11-29
> > > > > **Response to reviewer**
> > > > >
> > > > > We really appreciate the comments from the reviewer.
> > > > >
> > > > > ***Q1: Justification for the test design.*** \
> > > > > One of the reasons that we conduct the hypothesis test treating the average performance of a model on a dataset as a sample is that this setting satisfies the i.i.d assumption for the hypothesis test procedure. Testing on individual domains violates the i.i.d assumption in two ways: (i) independence is violated because for each performance measurement within the same dataset, there is significant overlap in the training datasets used to construct the model, so the performances of these models are correlated—this is known to cause issues when applying hypothesis tests to “standard” ML algorithms with accuracies measured using cross validation [A]; and (ii) the identically distributed assumption is violated because the performance of methods on domains in one dataset does not necessarily follow the same distribution as the performance of domains in another dataset. In contrast, doing the test over average accuracies does not violate either of these assumptions: no data points occur in more than one dataset (so independence is satisfied), and in the standard meta-learning setup one assumes that (in this case DG) tasks/datasets are IID samples from some distribution over tasks (so all dataset-level accuracies come from the same distribution).
> > > > >
> > > > > That said, we agree that the practical margin of ITL > Coral is small, even if it’s statistically significant (i.e. effect size vs significance). We will update the paper to acknowledge this. Investigating whether ITL can be complementary with orthogonal approaches to DG like CORAL is an interesting question for future work, as we have alluded to in our revised limitations paragraph.
> > > > >
> > > > > ***Q2: Justification for the test design.*** \
> > > > > Sorry for the confusion—the null hypothesis is incorrectly stated in the paper, but we will fix this. To clarify: We performed a one-sided hypothesis test with a significance level of 0.025, and use the following hypotheses: \
> > > > > H0: the performance of ITL-NET is less than or equal to ERM \
> > > > > H1: the performance of ITL-NET is greater than ERM
> > > > >
> > > > > ***Q3: Discrepancies.*** \
> > > > > Thanks for pointing out these editorial errors. In Table 6, there was a transcription error when copying the results from the DomainBed paper [B], resulting in the wrong number for ERM on V-Pascal which should be 74.6, and for CDANN on Artistic which should be 61.5. After replacing these two numbers, the average accuracy match exactly what we report in Table 6 and Table 2.
> > > > >
> > > > > ***Q4: ERM vs Coral in Tab 7.*** \
> > > > > Thanks for pointing this out. In the single source domain setting, ERM and CORAL essentially become the same algorithm (since coral is based on aligning multiple sources). So we could simply remove CORAL as there is not a meaningful difference.
> > > > >
> > > > > [A] Nadeau, Claude, and Bengio, Yoshua. Inference for the Generalisation Error. Machine Learning, 2003. \
> > > > > [B] Gulrajani, Ishaan, and David Lopez-Paz. "In Search of Lost Domain Generalization." International Conference on Learning Representations. 2020.

---

> > ### Comment · Reviewer_57HS · 2021-11-18
> > **Re: Typesetting**
> >
> > > The font size in the figures is increased and the highlight problem in the table captions has been solved. If you have a further concern about the format of the paper, please let us know, we will fix it.
> >
> > Thanks. The following concerns remain:
> >
> > - Figure 2 is still hard to read / seems to be strechted. I suggest re-plotting this.
> > - Font size in Figure 3 and Figure 4 is still unreadable.
> > - Table 1 caption currently has a typo: "Table 1: 0.8Resnet18"
> > - Remove the vertical lines from all tables --- cf. ICLR formatting instructions.
> >
> > I assigned my score independent of typesetting issues, but I still strongly encourage you to improve the paper plots to make an overall better impression with your paper. I suggest printing it to see the issue with the figures more clearly.

---

> > > ### Author Response · Authors · 2021-11-23
> > > **Response to reviewer**
> > >
> > > **The Typo and Figure problems** \
> > > Thanks for your suggestions. We have submitted a new revision where we have increased font size in Figure 3 and Figure 4 and also re-plotted Figure 2. The vertical lines are removed from all the tables. The typo in Table 1 is fixed. Please let us know any further concerns, we will update them in the camera-ready.

---

> > > > ### Comment · Reviewer_57HS · 2021-11-27
> > > > **Re: Re: Typesetting**
> > > >
> > > > Thanks for the update. While I still not particularly like the plotting style and quality, I think this is fine and readable now. Thanks for the update, it adresses my concern and meets the bar.
> > > >
> > > > [Some examples for further improvements: Why draw borders around legends? Legends overlap with plots. Legends are redundant (4x same legend, while nothing changes). Text is overlapping in the plots. The ticks are very small. Problematic to read in B&W, why not make one of the lines dashed (on top of color)? Text in Figure 2 is still streched, etc. While I think some of this is nitpicking or a matter of taste, it might be an easy way to please the reader here by additional improvements until the camera ready.]
> > > >
> > > > But to be clear, no further need for action here that would influence my perception of the paper and score, these are truly just some additional suggestions for the camera ready. Thanks!

---

> > ### Comment · Reviewer_57HS · 2021-11-18
> > **Re: Response to reviewer**
> >
> > Thanks for the response.
> >
> > You adressed the major points. Please check my comment about Table 2. Besides, I like the new comparison in Table 5, and the overview figure. I need to go through the full text again to check whether my other concerns were sufficiently addressed, but will reconsider my score afterwards.

---

> ### Comment · Reviewer_57HS · 2021-12-07
> **Post Rebuttal comments**
>
> Dear authors,
>
> I added my post-rebuttal comments in the main review summary and decided to increase my score to (6). Please let me know if you agree with the proposed changes or have final thoughts on the discussion around the statistics.

---

### Official Review · Reviewer_DuTo · 2021-11-05

**Correctness:** 3
**Technical Novelty And Significance:** 2
**Empirical Novelty And Significance:** 2
**Recommendation:** 6
**Confidence:** 4

**Main Review:**

+ The idea is interesting.
+ Lots of experiments and ablation is done.
---------------
- The paper lacks an ablation over multi-label binary sigmoid cross-entropy loss. The proposed loss is actually multi-label loss which regards the multi-class problem as a multi-label problem although the label vectors are always in the form of one-hot vectors. Multi-class cross-entropy is a multi-class loss. So it's more fair to compare the proposed loss with multi-label loss, instead of the multi-class cross-entropy loss, as the multi-label loss might be a good baseline for DG.
- The performances in DG is not SOTA and the paper lacks important references:
Self-Challenging Improves Cross-Domain Generalization (ECCV 2020) provides better performance in PACS while authors did not mention this paper in their paper. I doubt this paper can achieve SOTA performances in all the benchmarks.
- The writing and clarity can be improved in the following aspects:
a. the texts in fig. 1, 3, 4 are extremely small. It's better to enlarge them
b. the authors should add more descriptions about how they conducted analysis and experiments. For example, what is "+90%, +80%, - 90%" in table 5.



**Summary Of The Paper:**

This paper proposes a loss for domain generalization. The loss is learned by meta-learning on based on RotatedMNIST. The paper evaluate the loss's generalization ability in several other datasets.

**Summary Of The Review:**

This paper lacks evidences to show its SOTA performances and lacks important references and ablation study. I suggest authors to improve from the above points.

---

> ### Author Response · Authors · 2021-11-18
> **Response to reviewer**
>
> **Q1: Multi-label binary sigmoid cross-entropy loss** \
> ITL is not a multi-label loss. Please note that all three losses under discussion (i. conventional (multi-class) cross-entropy loss, ii. multi-label binary cross-entropy loss, iii. our ITL) de-compose as a sum over output units where each term in the sum depends on the unit’s prediction and true label, e.g., compare the sum over categories in Eq 8 in our revision to the same sum over categories in https://ml-cheatsheet.readthedocs.io/en/latest/loss_functions.html#cross-entropy). The more salient difference between multi-class and multi-label predictions is instead the activation used on the output layer of the network. Multi-class assumption uses softmax output to enforce sum to 1, and multi-label assumption uses a set of sigmoid outputs. All our problems are multi-class problems. So ITL, multi-class CE, and all other competitors in the initial submission use softmax activated output, making them multi-class losses. Therefore ITL does NOT make a different assumption to cross-entropy as claimed by the reviewer.
>
> Nevertheless, for a thorough evaluation, we also compared Sigmoid+BCE as a multi-label baseline (see revised Tab 1). The performance is unsurprisingly poor because the benchmark problems are not multi-label problems.
>
> **Q2: Comparison to RSC** \
> Self-Challenging Improves Cross-Domain Generalization (RSC) does have better self-reported results on PACS compared with ITL-Net. However, their results do not seem to not be reproducible. According to issues raised on the official RSC Github, many researchers have failed to replicate RSC results using their official code. We also re-ran their code and our results are similar to those reported by others A few percentage worse than the self reported results in RSC paper (see revised Tab 1). We also note that RSC did not report results on DomainBed, which should be consider the most reliable evaluation benchmark. We have now run the RSC provided code on DomainBed (see revised Tab 3), and it performs poorly when fairly compared under DomainBed’s standardized hyperparameter tuning protocol. Therefore overall we do not consider RSC to outperform ITL.
>
> More importantly, we emphasize that ITL is not a direct competitor to RS but to cross-entropy. RSC contributes some architectures and optimization algorithm innovations, but is agnostic to loss function. Original RSC used cross-entropy loss. We re-ran RSC’s provided code using ITL loss function in place of CE, and found a ~0.8% increase in performance (see Tab 1). Thus RSC and ITL are complementary.
>
> **Link:** https://github.com/DeLightCMU/RSC/issues/12
>
> **Q3: Font size in Fig.1,3,4 and caption in Table 5** \
> Thanks, we fixed the font size. The labels in Table 5 corresponded to the held-out domain descriptions in ColoredMNIST (colour perturbation). Please see [A] for further details. Anyway, we have now replaced this with a different dataset on suggestion of another reviewer (see revised Tab 5). Re: Experiment details, please let us know any part that you found ambiguous and we will address it.
>
> **Reference** \
> [A] Arjovsky, Martin, et al. “Invariant Risk Minimization.” ArXiv Preprint ArXiv:1907.02893, 2019.

---

> > ### Comment · Reviewer_DuTo · 2021-11-29
> > **Re: Response to reviewer**
> >
> > Thanks for the clarification. I think my concerns are addressed. After seeing other reviewers' comments, I think learning a proper loss for domain generation is an interesting idea. So I raise my score to 6.

---

### Author Response · Authors · 2021-11-18
**Revision uploaded**

We thank all the reviewers for their comments, and we have uploaded the updated manuscript according to the suggestions from the reviewers, where the updated parts are highlighted in Blue. In summary, our updates are as follows: \
**[DuTO]** BCE loss baseline is added Table 1. \
**[DuTO]** A comparison with RSC is presented in both Table 1 and Table 3. \
**[57HS]** An algorithm diagram is posted in Figure 1. \
**[57HS]** A discussion on the limits of our algorithm is added at the end of Section 4. \
**[57HS]** For repeatability analysis, we replaced CMNIST with OfficeHome. \
**[eX71]** We generated a learning curve of the meta-train stage to show the convergence behaviour of our inner loop and outer loop optimisation in Fig 6/ Appendix A.7. \
**[Hx46]** A short description for gradient-surgery is added. \
**[Hx46]** We add the sample size for the hypothesis testing in Table 3.

---

### Author Response · Authors · 2021-11-30
**General Response to Reviewers**

We thank the reviewers for the time they have put into reviewing the paper and their helpful feedback. In general, the reviewers all find our novel approach of learning a loss function to improve DG interesting and are impressed by the thorough empirical validation of the idea.

The main weaknesses identified by the reviewers stemmed from a lack of clarity at some points in the paper. We have since addressed this by changing some of the language, modifying/adding figures, and including a limitations section.

In light of this, we would appreciate it if the reviewers would reconsider their final scores.

---

### Decision · Program_Chairs · 2022-01-20

**Decision:**

Reject

**Comment:**

This paper considers the idea of meta-learning the loss function for domain generalization. It's a simple idea, that seems to work reasonably well. Although, as pointed out by the reviewers, the margin is actually quite modest when compared to the strongest baselines (not ERM).

On a positive note, many reviewers agree that the idea was simple, novel, and interesting. The insight that cross-entropy can be improved for domain generalization is interesting. On the other hand, many reviewers pointed out that the, despite some careful empirical work, it's not clear why this idea works. I read the paper myself, and I agree that the paper could use a bit more work before it is ready for publication. Specifically, I agree with Reviewer eZ71, who asked for a clear justification of the proposed idea. The idea seems sensible, but there is some burden on the paper to provide insight, and not simply present an idea.  Here are some specific suggestions that came up during discussion, which could strengthen the paper:
- A more comprehensive discussion of the limitations of this approach.
- It would be good to understand how critical was the specific choice of parametric loss family. Here are some questions that would be good to address: does the parametric family interact with the type of domain shift in the datasets? Why are Taylor polynomials preferable or beneficial for domain generalization compared to, e.g., a linear combination of standard loss functions?
- Is the dataset on which you learn your ITL loss critical? I.e., how critical was the choice of rotated MNIST for learning the ITL loss? Does it generalize to very different and more diverse domain shift tasks, like those in the WILDS benchmark? It would be particularly interesting to see if loss functions meta-trained on distinct datasets learn similar parameters.
- More broadly, evaluation on larger and more diverse domain shift tasks, like those in the WILDS benchmark, would further strengthen the conclusions in the paper.